# FUS unveiled in mitochondrial DNA repair and targeted ligase-1 expression rescues repair-defects in FUS-linked motor neuron disease

Manohar Kodavati [1,11], Haibo Wang [1,11], Wenting Guo[2,3,4], Joy Mitra[1], Pavana M. Hegde [1], Vincent Provasek[1,5], Vikas H. Maloji Rao[1], Indira Vedula[6], Aijun Zhang[6,7], Sankar Mitra[1], Alan E. Tomkinson [8], Dale J. Hamilton [6,7], Ludo Van Den Bosch[2,9] & Muralidhar L. Hegde [1,10] ✉

This study establishes the physiological role of Fused in Sarcoma (FUS) in mitochondrial DNA (mtDNA) repair and highlights its implications to the pathogenesis of FUS-associated neurodegenerative diseases such as amyotrophic lateral sclerosis (ALS). Endogenous FUS interacts with and recruits mtDNA Ligase IIIα (mtLig3) to DNA damage sites within mitochondria, a relationship essential for maintaining mtDNA repair and integrity in healthy cells. Using ALS patient-derived FUS mutant cell lines, a transgenic mouse model, and human autopsy samples, we discovered that compromised FUS functionality hinders mtLig3's repair role, resulting in increased mtDNA damage and mutations. These alterations cause various manifestations of mitochondrial dysfunction, particularly under stress conditions relevant to disease pathology. Importantly, rectifying FUS mutations in patient-derived induced pluripotent cells (iPSCs) preserves mtDNA integrity. Similarly, targeted introduction of human DNA Ligase 1 restores repair mechanisms and mitochondrial activity in FUS mutant cells, suggesting a potential therapeutic approach. Our findings unveil FUS's critical role in mitochondrial health and mtDNA repair, offering valuable insights into the mechanisms underlying mitochondrial dysfunction in FUS-associated motor neuron disease.

Fused in Sarcoma (FUS) is an important RNA/DNA binding protein involved in cellular metabolism, RNA processing, and DNA repair. It has been linked to neurodegenerative diseases, including amyotrophic lateral sclerosis (ALS) and frontotemporal dementia (FTD)[1–5]. FUS mutations have been identified, contributing to a significant proportion of familial and sporadic ALS cases[6,7]. While FUS mutations can disrupt gene transcription, mRNA splicing, and RNA transport, growing evidence suggests that DNA damage accumulation and repair deficiency also play critical roles in FUS-associated

neurodegeneration[8–11]. Studies have documented increased DNA damage in patient tissues and animal models with FUS pathology. Moreover, FUS interacts with various proteins such as XRCC1, PARP1, and HDAC1[9,12–21], participating in DNA damage repair processes. Recent research has highlighted the importance of FUS dependent liquid-liquid phase separation in initiating DNA repair[22].

In our previous investigation, we discovered that FUS forms a complex with PARP1, XRCC1, and DNA ligase IIIα (LigIII/Lig3) to initiate the repair of oxidative DNA damage and single-strand breaks (SSBs) in

the nuclear genome[22]. FUS plays a role in recruiting and enhancing the break-sealing activity of nuclear Lig3 (nuLig3) in a PARP activity-dependent fashion. While there are two other DNA ligases in the nucleus, namely Lig1 associated with replicating DNA and Lig4 linked to DNA double-strand break repair, mitochondrial version of Lig3 (mtLig3) is the only DNA ligase found in mammalian mitochondria and is involved in both mitochondrial genome repair and replication[23–25]. Unlike nuLig3, mtLig3 functions independently of XRCC1 and is essential for cell survival[23,26,27]. The non-essentiality of nuLig3 may be due to the presence of Lig1, which may act as a backup for nuLig3 deficiency in cycling cells[28]. Both mtLig3 and nuLig3 are generated by alternative translation initiation, resulting in the presence of N-terminal mitochondrial target sequence (MTS) in mtLig3. Thus, mtLig3, which has extensive sequence homology to nuLig3, is expected to interact with FUS. However, although overexpressed FUS has been shown to localize in mitochondria and abnormalities/damages to mitochondria are observed in FUS-ALS and FUS-FTD[29–31], its specific functions in mitochondria, particularly with respect to mitochondrial genome maintenance, and its overall native physiological role in mitochondria remain largely uncharacterized.

This study aims to investigate the role of endogenous wildtype (WT) FUS in mtDNA repair and elucidate the mechanisms by which FUS mutations contribute to mtDNA damage and neurodegeneration. We conducted a series of experiments using multiple cell lines including CRSPR/Cas9 mediated FUS knockout (KO) HEK293, first to establish the localization and DNA repair function of FUS in mitochondria. Subsequently, we investigated various disease-relevant models with FUS mutations/proteinopathy, such as ALS patient-derived cell lines with FUS mutations, their mutation-corrected isogenic control lines, autopsied ALS patient spinal cord tissues, as well as a human FUS R495X (hFUS*R495X) transgenic mouse model to uncover the implications of compromised FUS functions in causing mtDNA damage and overall mitochondrial dysfunction in FUS-associated ALS. Our findings reveal the crucial involvement of FUS in DNA strand break ligation activity by mtLig3. FUS is essential for the recruitment of mtLig3 at the mtDNA damage sites. The mtDNA repair defects observed in FUS KO and mutant cells correlated with the accumulation of damage and mutations in mtDNA, decreased mitochondrial membrane potential, and oxygen consumption rate (OCR), as revealed by DNA integrity assays, membrane potential tests, mtDNA sequencing, and Seahorse XFe96 analyses. Furthermore, correction of the FUS mutation by CRISPR/Cas9 mediated knock-in, or the targeted expression of an alternative DNA ligase (Lig1) restored mtDNA fidelity and functions, suggesting a potential for Lig1 in treating FUS associated neurodegeneration. In summary, our study uncovers the critical role of FUS in mtDNA repair and provides new insights into the mechanisms of mtDNA damage and repair defects in FUS-associated neurodegenerative diseases. It highlights the potential for therapeutic strategies aimed at correcting mtDNA repair defects and restoring mitochondrial function. This knowledge not only expands our understanding of the normal role of FUS in mitochondrial DNA repair and maintenance, but also opens avenues for further exploration of the pathogenesis of FUS-related neurodegeneration and developing targeted therapies.

## Results

### Localization of WT and mutant FUS and their link to mitochondrial abnormalities

Previous studies have reported the localization of FUS in mitochondria, but these observations were based on the overexpression of exogenous FUS[29,32]. In the current study, we first examined the endogenous levels of FUS in the mitochondria in HEK 293 cells and found that FUS levels were increased after induction of oxidative stress by glucose oxidase (GO) treatment that promotes oxidative DNA damage (Fig. 1a). To demonstrate the purity of the mitochondrial fraction, immunoblotting (IB) was performed using appropriate subcellular

fraction markers, namely proliferating cell nuclear antigen (PCNA) and GAPDH. The specificity of the FUS antibody was confirmed as it showed no signal in extracts from KO cells (Fig. 1a). We then measured mitochondrial FUS protein levels in cell lines derived from ALS patients. These cell lines included motor neurons differentiated from iPSCs and fibroblasts carrying either the WT or P525L mutant form of FUS. Our findings revealed that both WT and mutant FUS were present in the mitochondrial fraction. Notably, there was an increased mitochondrial localization of the P525L mutant compared to the WT form. Furthermore, upon treatment with GO, there was a moderate increase in FUS levels observed in both WT and mutant samples (Fig. 1b and Supplementary Fig. 1a). Furthermore, we employed a proximity ligation assay (PLA) to visualize the association of FUS and mitochondria using mitochondria-specific makers HSP60 and Tom20 in WT and mutant motor neurons and fibroblasts[33]. The results shown in Fig. 1c confirmed the association between FUS and the mitochondrial markers, where the mutant FUS showed a significantly higher count of PLA foci in comparison to the WT FUS. Magnified images related to Fig. 1c are shown in Supplementary Fig. 1b for a more detailed view. Interestingly, FUS P525L cells showed a higher number of PLA foci per cell compared to FUS R521H cells, which could be related to the degree of FUS mislocalization caused by different mutations (Supplementary Fig. 1c). In addition, the overall number of foci were enhanced by GO treatment in both FUS WT and mutant cells (Fig. 1d; magnified images in Supplementary Fig. 1b). We employed a PLA to further validate the mitochondrial association of endogenous FUS in iPSC-derived motor neurons. Figure 1e shows the presence of positive PLA foci between FUS and TOM20 in motor neurons under both untreated and GO-treated conditions. Notably, GO treatment appeared to enhance the formation of these PLA foci. MitoTracker green was used as a counterstain for mitochondria. The specificity of the antibodies used for PLA was rigorously tested in FUS KO cells, where no PLA foci was observed for FUS versus TOM20, reaffirming the specificity of the interaction. Mitochondria in these cells were also counterstained with Mito Tracker green, as shown in Supplementary Fig. 1d.

To assess mitochondrial function, we measured cellular respiration in fibroblasts (Fig. 2 and Supplementary Fig. 2) and neural lineage progenitor stem cells (NPSCs) differentiated from iPSCs (Supplementary Fig. 3a), both derived from ALS patients. These measurements were carried out using the Seahorse XF Cell Mito Stress Test protocol, both under baseline conditions and after the application of mitochondrial inhibitors. Our observations indicated no significant differences in OCR between untreated and GO-treated FUS WT and P525L mutant cells (Fig. 2a, b and Supplementary Fig. 3b, c)[34]. However, when assessing the recovery of FUS WT and P525L cells from GO-induced stress, we noted significant differences in non-mitochondrial oxygen consumption, and in their basal, maximal, and spare respiration capacities (Fig. 2c and Supplementary Fig. 3d). Notably, no significant differences were observed in proton leak and ATP production rates (Supplementary Fig. 2).

FUS has been shown to colocalize with stress granules (SGs) that have significantly accumulated in FUS mutant cells in response to sodium arsenite treatment, causing oxidative stress and protein misfolding leading to translation stalling[35]. This is believed to be a possible pathological mechanism of FUS aggregation formation in ALS patients. Given the localization of FUS in mitochondria and the colocalization of mutant FUS in SGs induced by sodium arsenite, we hypothesized that the accumulation of mutant FUS in SGs results in reduced FUS levels in mitochondria, thereby disturbing its normal function. Consistently, a decrease in FUS levels was observed in mitochondrial extracts from NPSC cells treated with sodium arsenite (Supplementary Fig. 2d). Furthermore, the OCR, as well as basal, maximal, and spare respiratory capacities of FUS P525L cells were markedly reduced compared to FUS WT after sodium arsenite treatment (Fig. 2d, e, and Supplementary Fig. 3e, f).

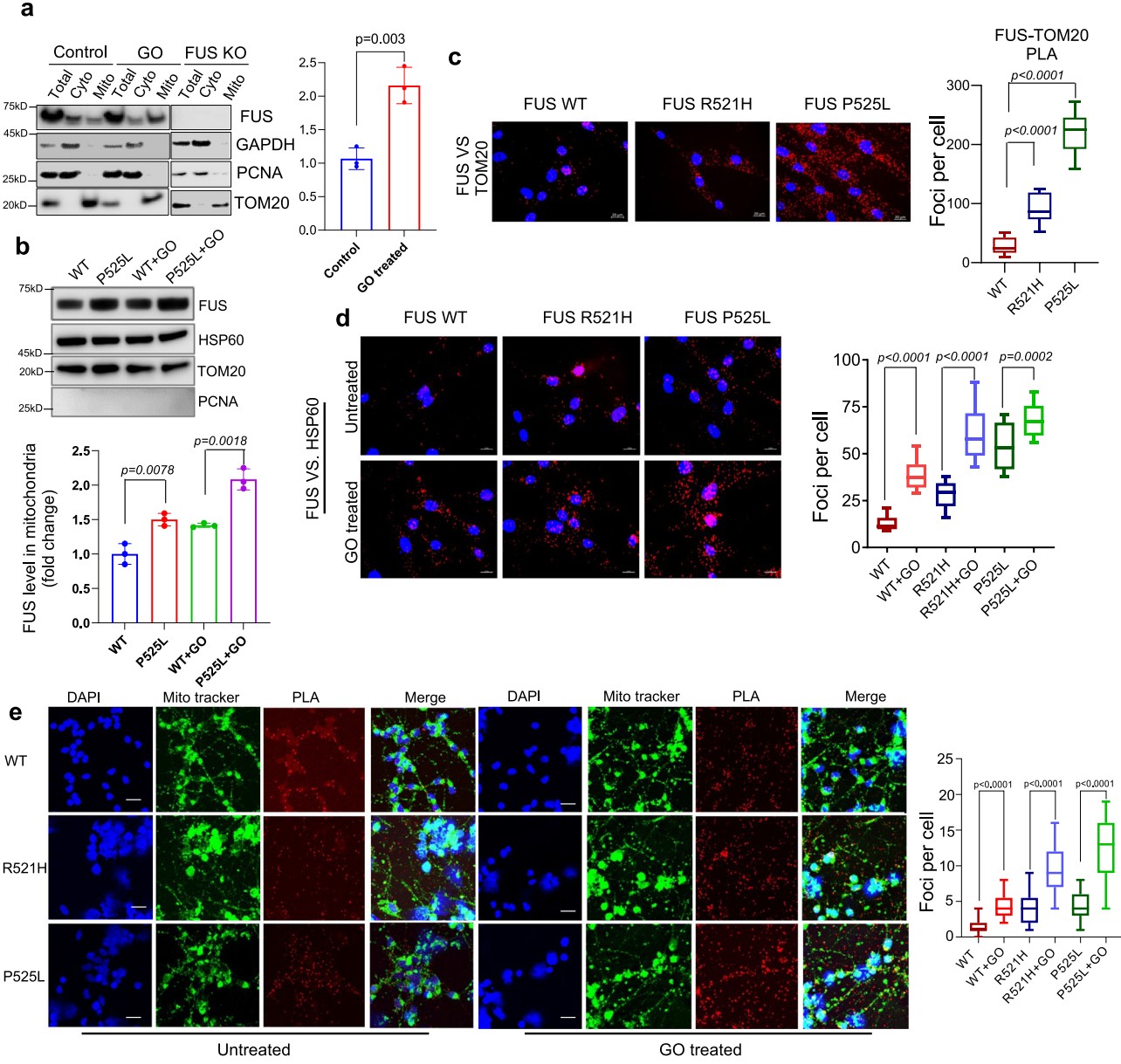

**Fig. 1 | Localization of endogenous wild type (WT) and mutant FUS to mitochondria: Both ALS associated FUS pathology or glucose oxidase stress increased mitochondrial localization. a, b** Immunoblots (IB) of cellular fractionization showing FUS in mitochondria. Total, cytoplasm and mitochondrial extracts are isolated from HEK 293 WT cells with and without glucose oxidase (GO) 100 ng/ml for 1 h and from FUS knockout (KO) cells in a. b corresponds to patient-derived iPSC differentiated into motor neurons with and without GO treatment, the data are presented as mean ± s.e.m. from three individual experiments, with the individual data points shown. **c–e** show proximity ligation assay (PLA) performed in patient derived fibroblasts and motor neurons, c corresponds to PLA of FUS with Tom20 without any stress along with quantifications. Scale bar = 20 μM. Box plot parameters: WT (minima 9, maxima 51, median 24, 25th percentile is 14 and 75th percentile is 39). For R521H (minima 51, maxima 125, median 89, 25th percentile is 115 and 75th percentile is 216). For P525L (minima 148, maxima 272, median 218, 25th percentile is 189 and 75th percentile is 239). **d** represents the comparison between PLA foci for FUS and HSP60 with and without GO and their quantifications. Box plot parameters: FUS WT (minima 8, maxima 21, median 12, 25th percentile is 10 and 75th percentile is 15). For FUS WT GO (minima 23, maxima 54, median 36, 25th percentile

is 31 and 75th percentile is 42). For FUS R521H (minima is 13, maxima 37, median is 27, 25th percentile is 21 and 75th percentile is 32). For R521H GO, (minima 41, maxima 79, median 56, 25th percentile is 47 and 75th percentile is 69). For FUS P525L, (minima 34, maxima 72, median 51, 25th percentile is 41 and 75th percentile is 65). For FUS P525L GO (minima 52, maxima 85, median 64, 25th percentile is 57 and 75th percentile is 74). **e** corresponds to PLA between FUS and TOM20 in motor neurons with and without GO treatment Nuclei stained with DAPI and mito-tracker green is used to counter stain mitochondria. Scale bar = 20 μM. The data are presented as mean ± s.e.m. from three individual experiments, quantification of PLA foci was derived from 25 cells. FUS WT (minima 0, maxima 4, median 1, 25th percentile is 1 and 75th percentile is 2), for FUS WT GO (minima 2, maxima 8, median 4, 25th percentile is 3 and 75th percentile is 5). For FUS R521H (minima 1, maxima 9, median 4, 25th percentile is 2 and 75th percentile is 5). For R521H GO (minima 5, maxima 16, median 9, 25th percentile is 7 and 75th percentile is 12). For FUS P525L (minima 1, maxima 8, median 4, 25th percentile is 3 and 75th percentile is 6). For FUS P525L GO (minima 4, maxima 19, median 13, 25th percentile is 9 and 75th percentile is 16). All statistical analysis were performed by two-sided student-t test using graph pad prism software. Source data are provided as a Source Data file.

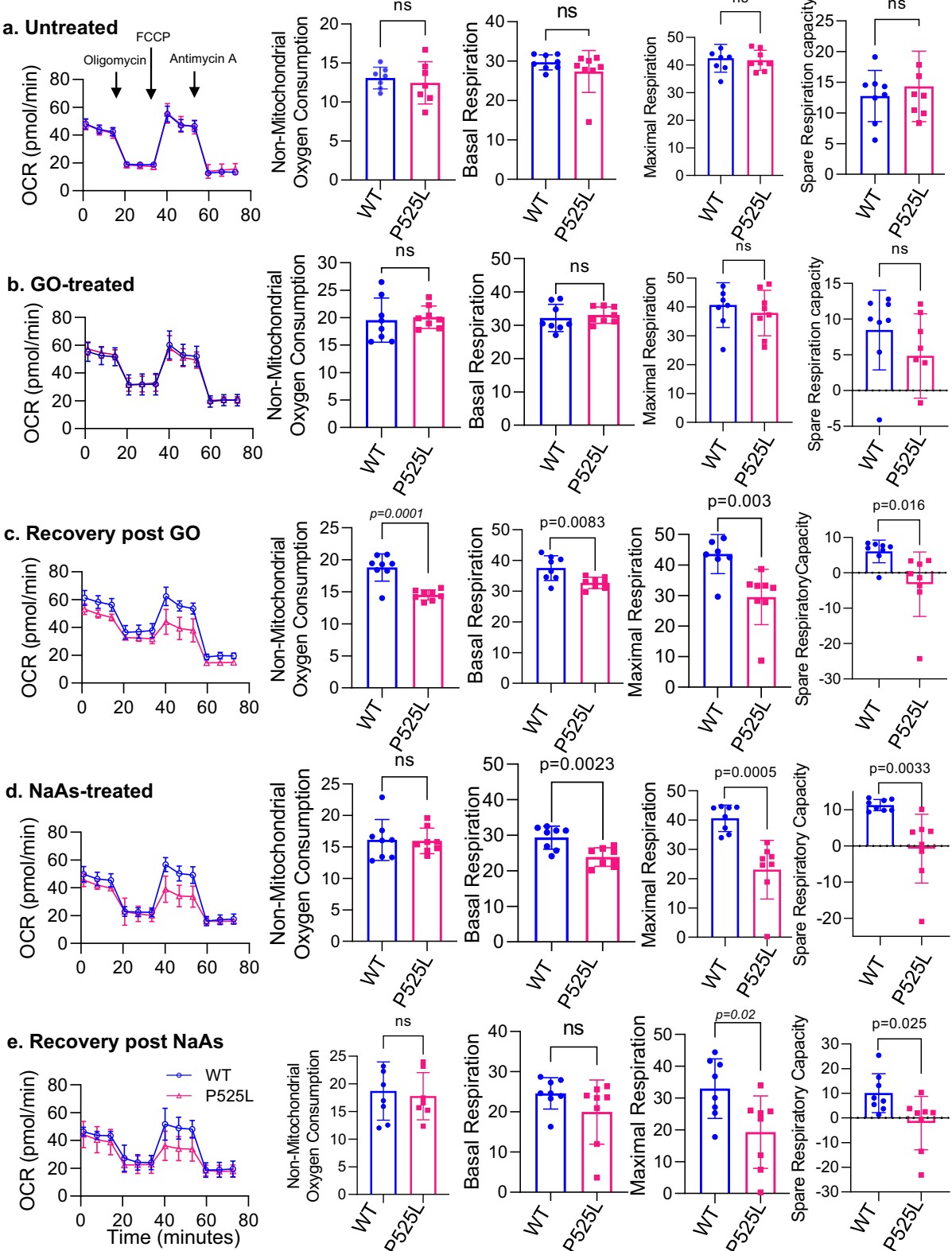

**Fig. 2 | Mitochondrial respiration comparison between FUS WT and FUS P525L patient derived fibroblasts by seahorse assay.** Untreated **a**, GO treated **b**, recovery after GO treatment **c**, sodium arsenite treated **d**, and recovery after sodium arsenite treatment **e**. Oxygen consumption rate (OCR) was determined throughout the mitochondrial respiration test in control and patient derived P525L fibroblasts. Arrows indicate the time when mitochondrial inhibitors were added to the media to assess respiratory parameters. Non-mitochondrial oxygen consumption was determined by measuring the difference between total oxygen consumption and antimycin A and rotenone treatment induced reduction in oxygen consumption, maximal respiration was expressed as difference between oxygen consumption following mitochondria uncoupling by FCCP and rotenone, antimycin A treatment and spare respiratory capacity was determined by subtracting basal respiration from maximal respiration. The data represent mean ± s.e.m. from three independent experiments and eight individual points are plotted in the graph. All statistical analysis were performed by two-sided student-t test using graph pad prism software. Source data are provided as a Source Data file.

Taken together, these results suggest that a portion of endogenous FUS is localized in mitochondria and that FUS depletion or mutations lead to alterations in both mitochondrial localization of FUS and mitochondrial function, such as mitochondrial membrane potential (MtMP). These findings are thus highly significant for understanding the role of FUS in mitochondria and its potential involvement in neurodegenerative diseases, such as ALS.

### DNA damage, mutation accumulation and repair deficiencies in mitochondria of ALS patient-derived FUS mutant cells, mouse model, and patient tissue

To assess the integrity of the mitochondrial genome, we used a LA-PCR method that detects DNA polymerase blocking lesions, which primarily include DNA strand breaks, cross-links or bulky base lesions[36]. To normalize the results, the mtDNA content and copy number were first determined by amplifying mtDNA relative to nuclear genes using the comparative Ct method (Supplementary Fig. 4a)[37]. To do this, real-time PCR was performed to amplify short fragment of three genes: *NADH-ubiquinone oxidoreductase chain 1 (ND1)* and *16SrRNA*, which are mitochondrial genes, *β2-microglobulin* for human, and *Hexokinase II(HK2)* for mouse, which are nuclear genes. This experiment was conducted in HEK293 FUS WT and KO cells first to identify the normal role of FUS in mitochondria of healthy cells, and then in mutant FUS iPSC-derived motor neurons and fibroblasts derived from ALS patients, and autopsy spinal cord tissues from FUS-ALS patients and hFUS*R495X mice[38], to unveil the implications in FUS-ALS.

LA-PCR was then performed in FUS WT and KO HEK293 cells by amplifying two long fragments ( > 8000 bp) of mtDNA (179-9231, and 7601-16407), and a short fragment ( ~ 200 bp) as a control. The results revealed a significant reduction in DNA integrity in FUS KO cells, which correlates with decreased membrane potential in FUS KO cells in comparison to WT cells, suggesting a FUS mediated effect on mitochondrial function (Fig. 3a and Supplementary Fig. 4b). A similar decrease in mtDNA integrity was observed in iPSC derived motor neurons and fibroblasts derived from ALS patients with FUS R521H and P525L mutations (Fig. 3b and Supplementary Fig. 4c, d). We also observed a significant decrease in mitochondrial (mt) DNA copy number in cell containing the FUS P525Lmutation compared to WT cells. This decrease in mt copy number correlated with a reduction in the expression of mitochondrially encoded genes, specifically Cytochrome B and ND4 (Supplementary Fig. 4c and e). These results are in line with previous findings that suggested a link between mutant FUS with a decrease in the levels of mitochondrially expressed proteins[39].

To further investigate the association between FUS and mtDNA integrity, we used a FUS transgenic mouse model expressing human FUS protein with the deletion of its nuclear localization signal (NLS)[38]. The mtDNA extracted from brains of hFUS*R495X mice showed decreased PCR products compared to WT mice, indicating the presence of strand breaks in the mtDNA of mutant mice (Fig. 3c and Supplementary Fig. 4f). We also observed an increase in mitochondrial localization of FUS in mice expressing FUS mutant, in comparison to control mice and mice overexpressing WT FUS (Supplementary Fig. 4g).

Finally, we conducted LA-PCR on mtDNA extracted from five ALS patient spinal cord tissue specimens with FUS pathology and three normal spinal cord samples as controls. The results showed significant accumulation of mtDNA damage in the patient samples (Fig. 3d and Supplementary Fig. 4h). The age of the patients and additional information can be found in Supplementary Table 1.

Previous research has established the involvement of FUS for optimal DNA damage repair. Building on this, our recent studies have uncovered the role of FUS in repairing DNA SSBs. We have demonstrated that FUS facilitates this repair process by enhancing the activities of DNA Lig3 in the nuclear genome[13]. Based on this evidence,

we hypothesized that the accumulation of DNA damage in mitochondria may be caused by a deficiency in DNA damage repair. To test this hypothesis, we evaluated the DNA damage repair kinetics in HEK293 FUS KO cells and patient derived FUS-mutant motor neurons and fibroblasts. HEK293 cells were treated with a DNA damaging agent (GO), and LA-PCR was performed 30 and 180 minutes after treatment to assess mtDNA damage repair. These results showed that while mtDNA integrity does recover in both FUS WT and FUS KO cells, loss of FUS caused reduced recovery after 2 h. One contributing factor for this observed difference may be due to a more pronounced initial decline in integrity in FUS KO cells at 0.5 h, suggesting an increased susceptibility to damage in these cells, in addition to possible impaired repair, Additionally, FUS R521H and P525L mutated cells showed a delayed DNA damage repair response during recovery from the GO treatment, normalized to the relative mtDNA content, at 30, 60, and 180 min (Fig. 3e, f and Supplementary Fig. 4i, j).

To further understand the possible connection between DNA damage repair defects and mtDNA instability, we performed mitochondrial DNA sequencing to measure insertions, deletions, and mutations in FUS WT, KO, and mutated cells as well as ALS patient spinal cord samples. Due to the challenges of obtaining pure and sufficient mtDNA, we performed PCR to amplify mtDNA before subjecting to sequencing analysis. We identified several unique mutations in FUS KO and mutant cells, as well as in patient tissues, and we evaluated the severity of each mutation using PolyPhen-2 online tool (http://genetics.bwh.harvard.edu/pph2/). Our results showed an increased mtDNA instability in these cells and tissues associated with FUS pathology, as indicated by the high number and severity of mutations (Fig. 3g). The types of novel mutations obtained from the sequencing data are listed in Supplementary Table 3.

In conclusion, our data show a significant reduction in mtDNA integrity in FUS KO cells, patient derived motor neurons, fibroblasts, spinal cord tissue specimens from ALS patients, and FUS transgenic mice. This suggests that the accumulation of DNA damage in mitochondria and a deficiency in DNA damage repair are associated with FUS pathology in ALS. Furthermore, our evaluation of DNA damage repair kinetics and mitochondrial DNA sequencing revealed a significant increase in mtDNA instability associated with FUS mutations. These findings revealed a direct link between FUS proteinopathy and mtDNA instability and led us to further explore the relationship between FUS mediated mtDNA instability and parameters of mitochondrial dysfunction in the disease.

### Impaired DNA ligation activity and recruitment of Lig3 to mtDNA damage sites in cells with FUS P525L mutation

Based on our previous study that showed the inhibition of nuclear Lig3 activity in FUS mutant cells, we investigated whether the impaired mtDNA damage repair observed in FUS mutant cells is due to reduced Lig3 activity in mitochondria. As we previously reported the association between FUS and Lig3 in response to oxidative stress in nuclei, we first examined the protein levels of Lig3 in mitochondrial extracts of FUS WT and FUS KO cells[13]. We found that the mtLig3 protein levels were not affected by FUS KO (Fig. 4a), and while XRCC1 was not detected, Lig3 was detectable in mitochondria, consistent with previous reports[26]. Furthermore, mtLig3 levels were slightly elevated in cells treated with GO (Fig. 4b), and the interaction between FUS and mtLig3 was increased in the presence of GO (Fig. 4c). To determine the effect of FUS mutations on their interaction with Lig3 within mitochondria, we performed a co-IP experiment using patient derived NPSCs. Our observations revealed a reduced interaction between FUS and Lig3 in FUS mutant cells compared to FUS WT, suggesting that the FUS-Lig3 interaction may be compromised in ALS FUS pathology involving FUS mutations (Fig. 4d).

To investigate the interaction between WT or mutant FUS and Lig3, we performed PLA in control and patient motor neurons and

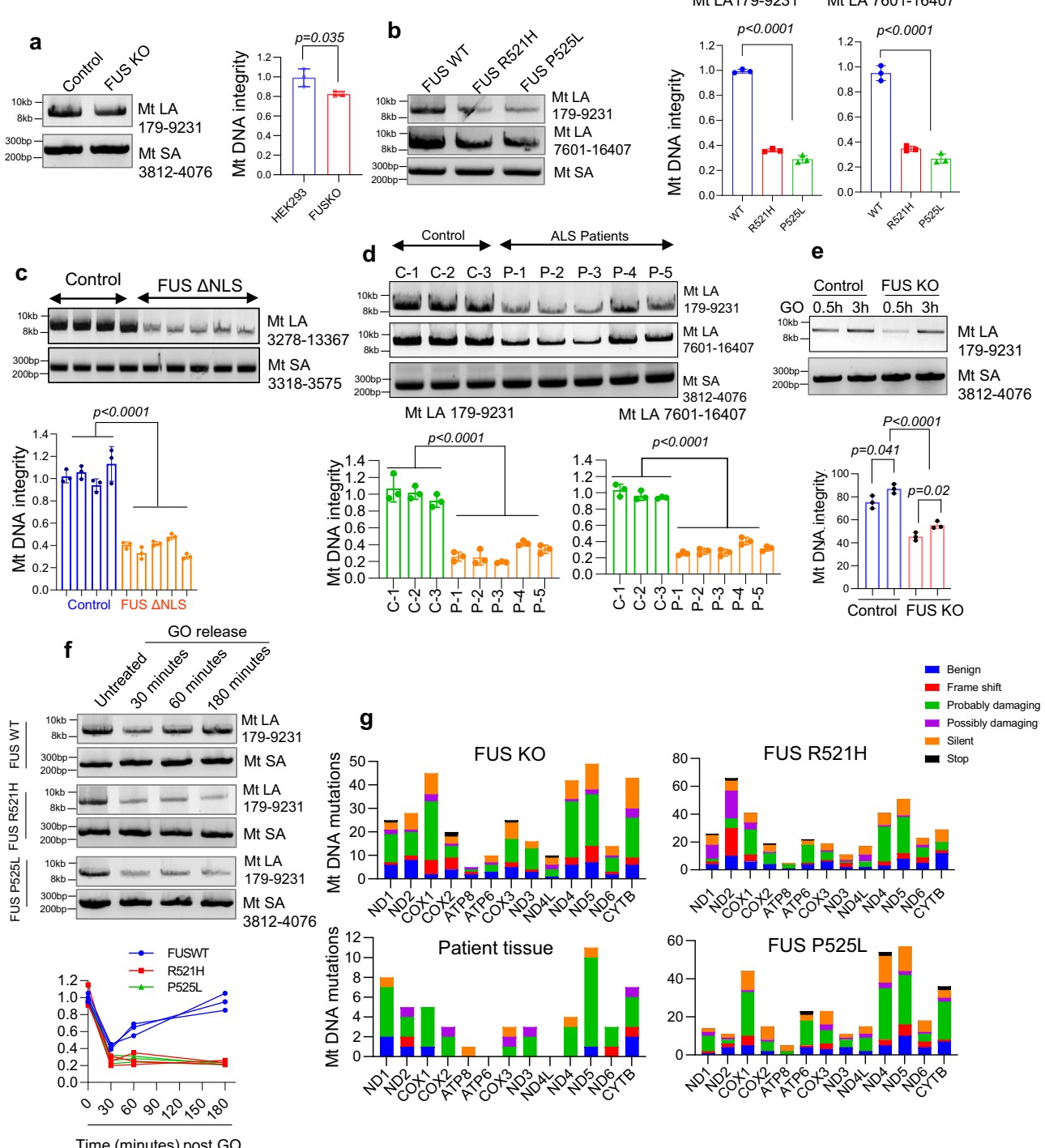

**Fig. 3 | Accumulation of DNA damage and deficient DNA damage repair in ALS patient-derived FUS mutant cells. a–f** shows long amplification PCR (LA-PCR) to determine genomic integrity of mt DNA, >8000 bp fragment of mtDNA was amplified and separated in 1% agarose gel along with a control PCR of 200 bp (MtSA). Amplified PCR product is quantified using pico green fluorescence, **a** shows comparison between the control and FUS KO HEK293 cells, **b** shows comparison of genomic integrity between control and patient derived mutant motor neurons using two amplicons. **c** represents the comparison of genomic integrity between WT and FUS ΔNLS mice brain tissue at around 12-month age, **d** corresponds to human spinal cord tissue samples between control and ALS patient spinal cord with FUS pathology. **e, f** differentiate the ability of FUS mutated cells and of FUS KO cells to repair oxidative DNA damage, **e** HEK 293 cells whereas **f** represents iPSC derived motor neurons. All LA-PCR data are presented as mean ± s.e.m. from three individual experiments, with the individual data points shown, all statistical analysis were performed using two-sided student-t test in graph pad prism software. **g** corresponds to mitochondrial DNA sequencing in HEK 293 FUS KO cells, patient spinal cord tissues and patient derived fibroblasts performed using mitochondrial REPLI-g kit. The unique mutations in protein coding genes are represented based on the kind of mutation and the severity. The frame shift represents either insertions or deletions. Silent mutations showed coding for same amino acid, and the severity of mutation was determined either benign, probably damaging and possibly damaging based on polyphen2 analysis. The Mitochondrial DNA sequencing was performed once using PCR amplified targeting mitochondrial DNA. Source data are provided as a Source Data file.

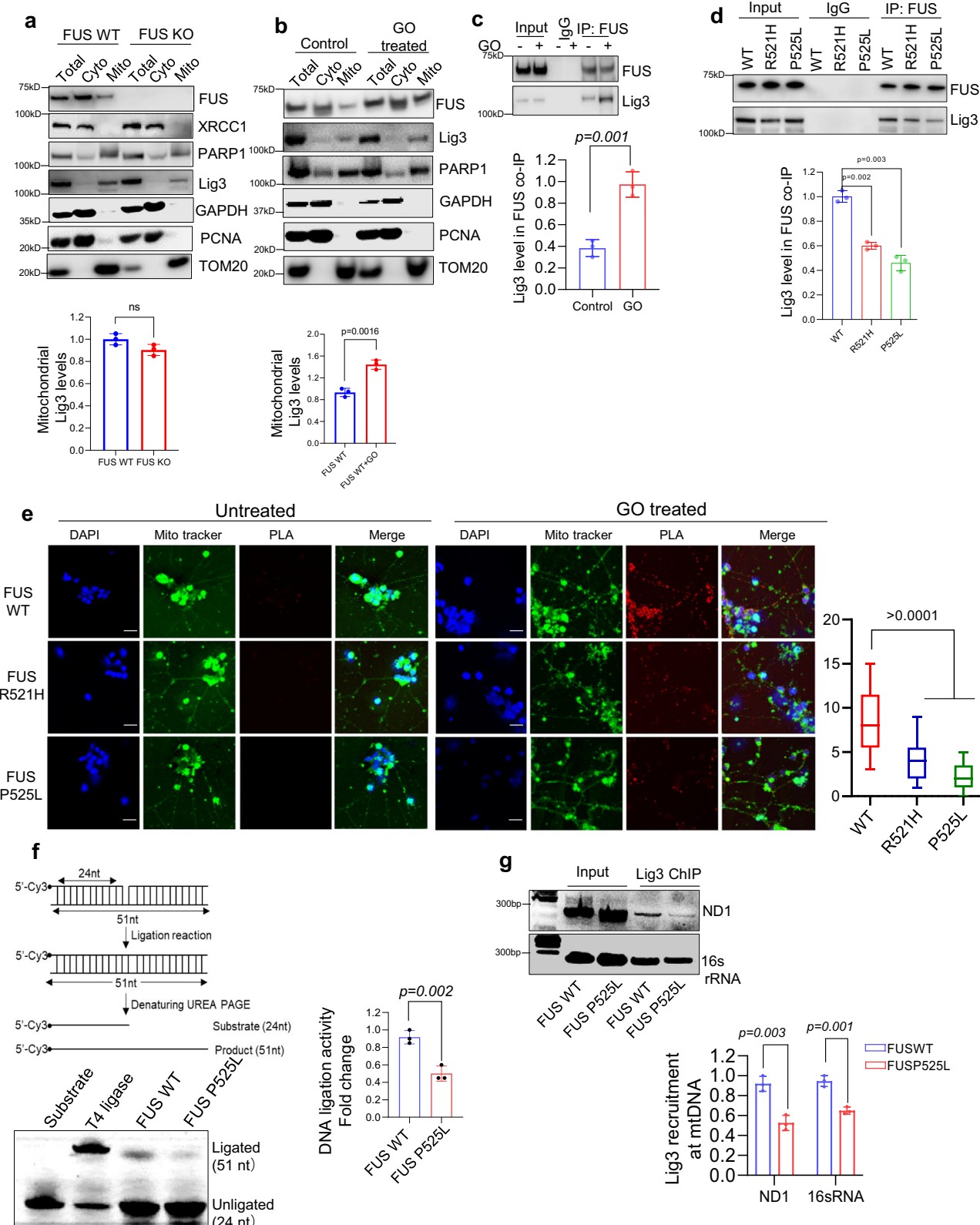

fibroblasts. As shown in (Fig. 4e; and Supplementary Fig. 5a), the association between FUS and Lig3 was significantly increased in response to GO treatment, but the interaction was significantly reduced in cells with mutant FUS compared to those with WT FUS (Fig. 4e and Supplementary Fig. 5a). We also compared the DNA ligation activity of Lig3 in mitochondrial extracts by an in vitro ligation activity assay. Although the mitochondrial DNA ligation activity was below the experimental detection level in untreated cells, it increased significantly in GO-treated cells. To compare the ligation activity between FUS P525L mutant and WT cells, we used mitochondrial total lysate from GO-treated patient derived cells. We observed a decrease in ligation activity in FUS mutant cells (Fig. 4f).

To investigate the possibility that the reduced ligation activity in FUS mutant cells was due to the failure of mtLig3 to be recruited to

**Fig. 4 | FUS mutation impairs the ligation activity as well as the recruitment of Lig3 to DNA damage sites. a, b** IB showing mitochondrial localization of key DNA repair proteins involved in oxidative damage response in HEK 293 WT and FUS KO cells mitochondria in a and WT cells treated with and without GO treatment (100 ng/ml for 1 h) in **b. c, d** Represent IB of endogenous FUS co-IP with Lig3, **c** corresponds to HEK293 cells and **d** corresponds to patient derived NPSC cells. The IP was performed with anti-FUS antibody, the data are presented as mean ± s.e.m. from three individual experiments, with the individual data points shown. **e** PLA of FUS vs Lig3 in control and patient derived motor neurons with or without GO treatment. Nuclei were stained with DAPI and mito-tracker green counter-stained mitochondria. Scale bar=20 µM. The box plot parameters: FUS WT GO (minima 3, maxima 15, median 8, 25th percentile is 5 and 75th percentile is 11). For R521H GO (minima 1, maxima 9, median 4, 25th percentile is 2 and 75th percentile is 5). and for FUS P525L GO (minima 0, maxima 5, median 2, 25th percentile is 1 and 75th percentile is 3). The data is derived from three individual experiments. Quantification of PLA foci from 25 motor neuron cells. **f** In vitro nick ligation activity assay performed using control and patient derived fibroblast mitochondrial extracts. **g** ChIP analysis of WT and FUSP525L fibroblast mitochondria with Lig3 antibody reveals reduced enrichment at mitochondrial DNA after GO treatment, data is derived from three individual experiments. All statistical analysis were performed using two-sided student-t test in graph pad prism software. Source data are provided as a Source Data file.

DNA damage sites, we conducted a chromatin immunoprecipitation (ChIP) assay. We amplified short fragments (~ 200 bp) of two mitochondria genes, *ND1* and *16 s rRNA*, in mtLig3 immunoprecipitant. We found significantly decreased PCR products in FUS mutant samples (Fig. 4g and Supplementary Fig. 5b), which indicates that mtLig3 was not optimally recruited to DNA damage sites due to the FUS mutation. Furthermore, as a positive control to validate the effectiveness of the ChIP assay, specifically to ascertain that the presence of mtDNA damage under our experimental conditions does not interfere with the ChIP product amplification, we performed a ChIP experiment in FUS WT NPSCs, both before and after GO treatment. Our results revealed a pronounced increase in Lig3 recruitment to mtDNA in the cells subjected to GO treatment, indicating a response to induced mtDNA damage (Supplementary Fig. 5c, d). Taken together, these results suggest that FUS mutation impairs both the ligation activity of mtLig3 and its recruitment to mtDNA damage sites, which could contribute to the observed mtDNA damage repair defects in mutant FUS cells.

## Role of FUS P525L mutation in mtDNA damage repair defects and its rescue by targeted expression of Lig1

To directly link the deficiency in mtDNA damage repair to the FUS mutation, we measured mtDNA repair in motor neurons differentiated from an isogenic iPSC line, in which the P525L mutation was corrected using the CRISPR/Cas9 system[31]. Immunofluorescence images showed that the cytoplasmic accumulation of mutant FUS was reduced in the isogenic control cells, closely resembling that of WT cells (Fig. 5a). Moreover, western blot analysis indicated that the total protein level of FUS was unaffected, while the mitochondrial FUS level was slightly decreased in the mutation-corrected isogenic control, likely due to the recovery of the cytoplasmic accumulation (Fig. 5b). We then assessed the DNA repair capacity by LA-PCR at different time points after releasing the cells from GO treatment. Both P525L mutant motor neurons and their isogenic control were treated with 100 ng/ml of GO for 1 h, followed by recovery for 30, 60, and 180 min, respectively. Compared to the mutant, the isogenic control showed a much higher DNA integrity at 180 min, which was comparable with FUS WT (Supplementary Fig. 6a).

These data suggest that the DNA damage repair deficiency observed in FUS mutated cells is caused by the P525L mutation. Therefore, we hypothesized that the ligation activity of mtLig3 in FUS P525L mutated cells could be rescued by the mitochondria-targeted expression of human DNA Lig1, another major DNA ligase specifically expressed in mammalian nuclei but not in mitochondria that is critical to DNA SSB repair, primarily in dividing cells[40]. Notably, Lig1 is minimally expressed in non-dividing cells, including postmitotic neurons, and the absence of Lig1 may worsen Lig3 defects in neurons more than in dividing cells. Additionally, the pathways involving Lig1 recruitment to DNA damage sites are different from those involving Lig3. Lig3 recruitment in nuclear genomes relies on the PARP1 mediated binding with XRCC1 to form a complex, and the ligation activity of Lig3 is promoted by FUS in both nucleus and mitochondria. Conversely, Lig1 requires interaction with PCNA for recruitment to DNA damage sites in nuclear genomes[40]. However, studies have shown that mitochondrial

targeting sequence (MTS) tagged Lig1 (MTS-Lig1) can localize to mitochondria in the absence of PCNA and provide ligase activity in Lig3 KO cells[27]. To target Lig1 to mitochondria, we cloned 22-amino acid MTS derived from the precursor of human cytochrome oxidase subunit 8 A (COX8) and added it to the N terminal of human Lig1 ORF following a FLAG tag in pcDNA plasmid (Fig. 5c)[41]. We verified the specific expression of this plasmid in mitochondria by Western blot (Fig. 5c) after transfecting it into patient derived fibroblast cells and immunofluorescence in HEK293 cells transfected with MTS-Lig1 plasmid (Supplementary Fig. 6b).

We first tested the overall ligation activity in mitochondria by in vitro ligation activity assay and found that MTS-Lig1expressing NPSC and fibroblasts with FUS P525L mutation had a significantly higher ligation activity than the control (Fig. 5d and Supplementary Fig. 6c). Next, we compared the oxidative DNA damage repair in MTS-Lig1expressing cells and controls by LA-PCR and observed that Lig1 enabled mitochondria to rescue the DNA damage repair deficiency induced by the FUS P525L mutation (Fig. 5e). We also measured mitochondrial membrane potentials in FUS WT and P525L mutant cells with or without MTS-Lig1 and observed improved membrane potentials in MTS-Lig1 expressing FUS P525L NPSC cells (Fig. 5f).

Overall, the results suggest that the mtDNA damage repair deficiency observed in FUS P525L mutated cells is linked to the mutation and that the reduced ligation activity of mtLig3 in these cells can be rescued by the expression of MTS-Lig1. This indicates that MTS-Lig1 has therapeutic potential for treating ALS caused by FUS mutations (Schematically shown in Fig. 6).

## Discussion

The instability of mtDNA is a hallmark of aging and neurodegeneration[42]. As individuals age, mutations accumulate in mtDNA, which leads to decreased mitochondrial function, cellular stress, and inflammation[43,44]. Neurodegenerative diseases such as Alzheimer's, Parkinson's, and ALS are also associated with mtDNA instability, which contributes to progressive neuron degeneration[45,46]. The accumulation of mtDNA mutations and impaired mitochondrial function in neurons and other brain cells, can lead to oxidative stress, inflammation, and energy deficits in the brain[47–49]. MtDNA damage can cause defective bioenergetics, reduced cell proliferation, and apoptosis. The mtDNA is considered more vulnerable to damage than nuclear DNA due to its proximity to the oxidative environment within mitochondria. The increased vulnerability to damage in mitochondria, as compared to the nucleus, may be partly due to the lack of protective histones and several DNA repair mechanisms that are present in the nucleus but absent in mitochondria[50]. Therefore, preserving mitochondrial function and mtDNA fidelity can potentially delay aging and treat neurodegenerative disorders.

Our studies show that the FUS protein, which is implicated in the etiology of both ALS and FTD, plays a crucial physiological role in repairing mtDNA. FUS was previously reported to localize in mitochondria[31]. However, these observations were based on the overexpression of exogenous FUS[29,32]. In this study, we examined the endogenous protein level of FUS in mitochondria in HEK293 cells and

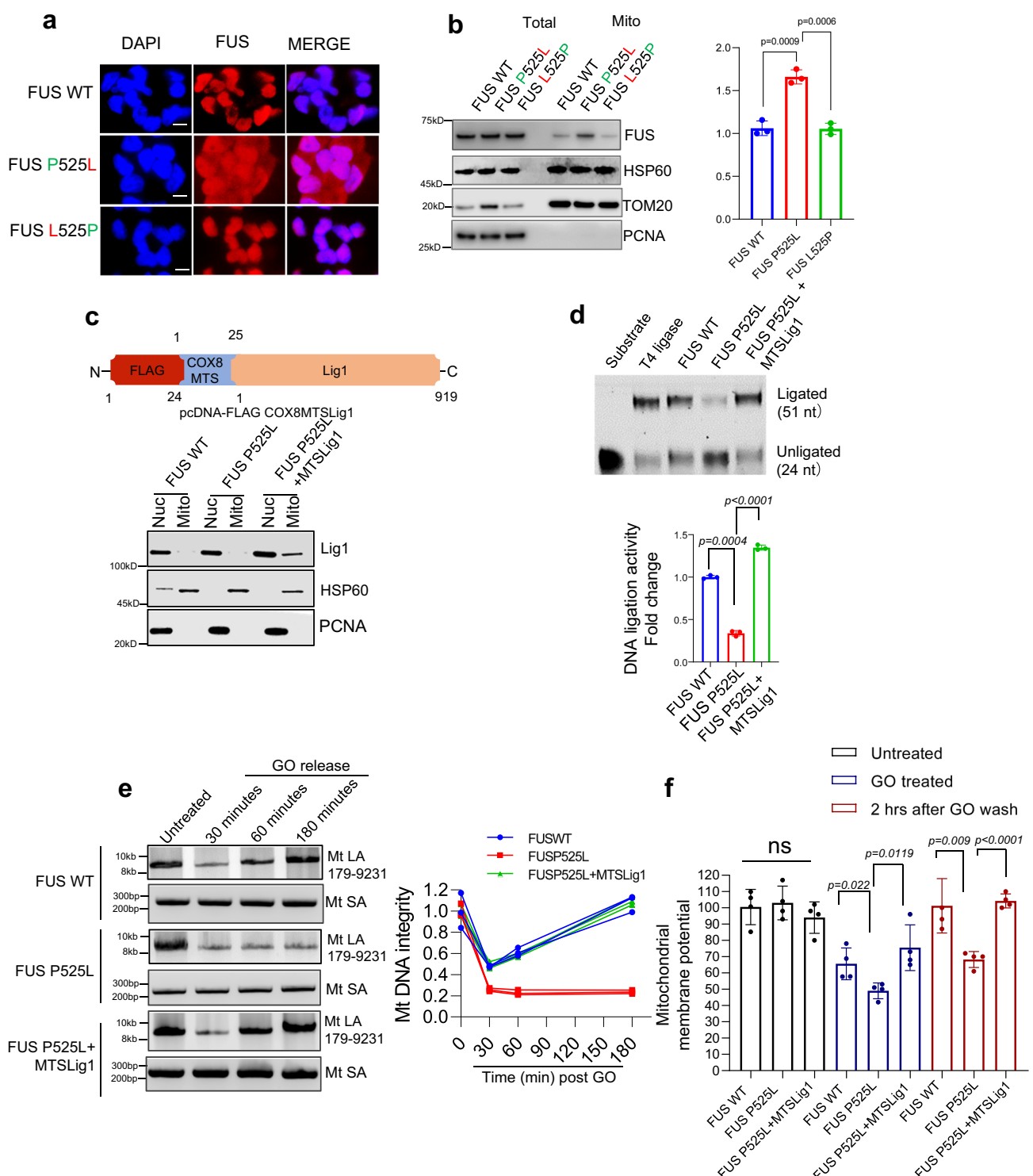

showed that mitochondrial localization of endogenous FUS is increased upon induction of oxidative stress by GO treatment. We also observed an interaction between FUS and mtLig3, which was enhanced by treatment with GO, suggesting a functional interaction between FUS and mtLig3. Our data from familial ALS patient-derived motor neurons and fibroblasts indicate that mutant FUS is recruited to mitochondria at a higher level than WT FUS, and that this recruitment is further increased upon DNA damage induction by GO exposure. However, mutant FUS is unable to proficiently interact with mtLig3 compared to WT FUS and this reduced interaction is responsible for reduced mtDNA Lig3 activities. Furthermore, we found that FUS

mutations cause defective recruitment of Lig3 to mtDNA, and that this leads to mtDNA damage and accumulation of mutations. Consistently, unlike WT FUS, increased mutant FUS in mitochondria results in DNA repair defects and mitochondrial dysfunction. Mutant FUS iPSC derived motor neurons also show a decrease in total mitochondrial content and mitochondrial motility. Additionally, we found that the presence of FUS is required for proper mitochondrial function, as demonstrated by reduction in mitochondrial membrane potential in FUS KO models and iPSC-derived motor neurons. In patient-derived cells with FUS P525L mutation, we observed reduced mitochondrial function recovery after GO and sodium arsenite treatment in

**Fig. 5 | Mitochondria DNA damage accumulation and DNA repair defects can be rescued by the correction of FUS mutations or targeted Lig1 expression. a** IF of endogenous FUS localization in control and patient derived motor neurons along with isogenic mutation corrected cells, IF was performed by anti-FUS antibody. DAPI staining indicates nucleus. Scale bar = 10 μm, the experiment was performed in three independent repetitions. **b** IB of endogenous FUS, HSP60, Tom20, and PCNA in FUS WT, FUS P525L and isogenic FUS L525P motor neurons, the data are presented as mean ± s.e.m. from three individual experiments, with the individual data points shown. **c** Human DNA ligase 1 (Lig1) localizing in mitochondria (FLAG-MTS) was generated with n-terminal FLAG and mitochondrial localization signal (MTS) from Cytochrome c oxidase subunit 8 (COX8) gene, IB showing Lig1expression in nucleus and mitochondria in patient derived fibroblasts, the experiment was performed in three independent repetitions. **d** In vitro nick ligation activity assay performed using patient derived NPSC mitochondrial extracts with MTS-Lig1 expression, the data are presented as mean ± s.e.m. from three individual experiments, with the individual data points shown. **e** LA-PCR-based DNA damage repair kinetic analysis. Genome DNA extracted from control and patient derived NPSC with FUS WT, FUS P525L and FUS P525L MTS-lig1 at indicated time points after release from exposure to GO (100 ng/ml) for 1 h. Amplification products analyzed by agarose gel electrophoresis and pico green-based quantitation represented, the LA-PCR data are presented as mean ± s.e.m. from three individual experiments, with the individual data points shown. **f** microplate reader-based analysis of TMRM signal intensity, comparison of control, FUS P525L and FUS P525L cells expressing MTS-Lig1. The cells were treated with GO (100 ng/ml) for 1 h and released for 2 h post 1 h treatment. the data are presented as mean ± s.e.m. from four individual experiments, with the individual data points shown. All statistical analysis were performed using two-sided student-t test in graph pad prism software. Source data are provided as a Source Data file.

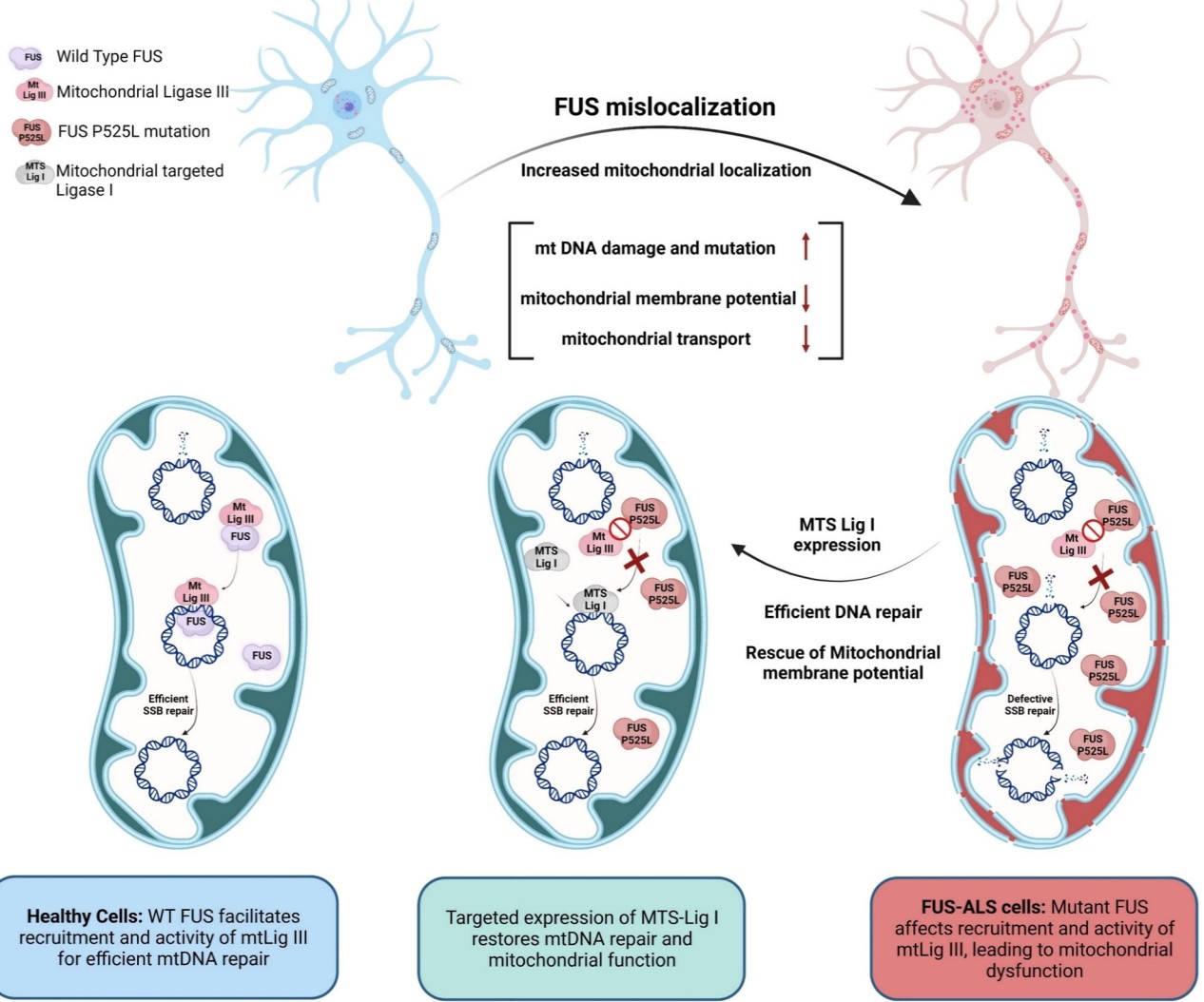

**Fig. 6 | A schematic model summarizing our findings on the role of mutant FUS in inducing mitochondrial dysfunction and mtDNA instability in ALS/FTD, and its amelioration by targeted expression of DNA Lig1.** The optimal FUS recruitment to mitochondria is critical for maintaining mtDNA integrity, as FUS promotes mtDNA Lig3 function through direct interaction, independent of XRCC1 in healthy neurons. However, ALS pathology associated FUS mutations lead to nuclear clearance and increased mitochondrial localization of FUS, and the mutant FUS fails to interact with mtLig3, causing defective recruitment to mtDNA damage sites and increased mutational load, ultimately resulting in mitochondrial dysfunction. The targeted expression of Lig1 in FUS-mutated cell mitochondria restores DNA repair and improves overall mitochondrial function. Created with BioRender.com.

comparison to control cells. To further investigate the effects of the mutation on mitochondrial function, we measured various mitochondrial functional parameters using mitochondrial stress test assay on Seahorse XFe96 platform. We found that mutant cells have defective recovery in various mitochondrial respiratory function parameters in comparison to WT cells after genotoxic and protein aggregation stress. We also observed increased mtDNA damage in spinal cord tissues from ALS patients with FUS pathology, as well as

decreased ability to repair damaged DNA in FUS mutant cells compared to WT cells. Our study identified DNA ligation defects associated with FUS pathology as a key driver of DNA damage and defective DNA repair in mitochondria. Furthermore, our results suggest that restoring mitochondrial ligase activity by expressing DNA Lig1 could potentially rescue genomic instability in mitochondria caused by FUS pathology.

The FUS P525L mutation, which is associated with a severe form of juvenile ALS (jALS)[51], was found to cause a substantial decrease in mtDNA ligase activity in cells carrying this mutation. The relationship between jALS and mtLig3 activity is an intriguing area warranting further investigation. Understanding the connection between this mutation and mtDNA instability could shed light on the underlying mechanisms of jALS and could lead to new therapeutic strategies.

The mechanisms of mtDNA repair rely on the translocation of nuclear-encoded DNA repair proteins. Recent studies have revealed varying degrees of recruitment of DNA repair proteins to the mitochondria. Notably, mtLig3 is the only DNA ligase that is recruited to mitochondria[50]. This is due to mtLig3 being an alternative splice variant of the Lig3 gene that retains a N-terminal MTS. XRCC1 is another critical protein involved in nuclear SSB repair[24,26,52]. It coordinates the functions of various reactions involved in DNA end processing, including the DNA nick sealing by Lig3[25]. However, XRCC1 is not recruited to mitochondria. In the absence of XRCC1, it is conceivable that FUS acts as a scaffolding factor for mtLig3.

Mutations in mtDNA have been implicated in ALS that can impair the function of the mitochondria, leading to decreased energy production, increased oxidative stress, and cellular stress[53-55]. This can contribute to the degeneration of motor neurons, which are highly reliant on mitochondrial function to maintain their energy demands. Targeting mtDNA mutations and preserving mitochondrial function could hold promise in delaying the progression of ALS and improving outcomes for affected individuals. However, the underlying mechanisms that generate these mutations are not well understood. Our mtDNA sequencing analyses reveal a substantial increase in mtDNA variations, including base changes, deletions, and insertions in cells affected by the mutant FUS protein. This provides a potential connection between mtDNA repair defects and the accumulation of mtDNA mutations in FUS-related neurodegeneration.

Overall, our study establishes two main outcomes. Firstly, we established the normal function of FUS in mtDNA repair using WT and FUS KO cells. Through these experiments, we directly demonstrated the crucial role of FUS in maintaining mtDNA integrity through its involvement in repair mechanisms.

Secondly, we uncovered the impact of FUS proteinopathy/mutations in causing mtDNA damage, repair defects, and broader mitochondrial dysfunction observed in FUS-associated neurodegeneration. By examining disease-relevant cell lines, mouse models, and autopsy tissues with FUS mutations, we demonstrated the detrimental consequences of compromised FUS functions for mitochondria. These findings add to the growing body of evidence linking mtDNA damage to neurodegenerative diseases.

Finally, we demonstrate that a targeted Lig1 expression offers a promising approach to restore mtDNA integrity and function. The ability to selectively deliver Lig1 to mitochondria to counteract mutant FUS induced Lig3 defects and to reduce the accumulation of mtDNA damage and prevent the onset and progression of mitochondrial dysfunction in FUS-associated neurodegeneration. Moreover, the high specificity and low toxicity of Lig1-based therapy make it a safe and effective treatment option. Thus, our findings, particularly the revelation of compromised mitochondrial DNA repair efficiency observed in ALS-motor neurons are clinically significant. We propose that this inefficiency in mitochondrial DNA repair mechanisms may play a pivotal role in the progression and exacerbation of neurodegeneration in FUS-ALS. The impairment in mitochondrial DNA repair could lead to increased neuronal vulnerability and accelerated cell death,

contributing to the rapid progression of ALS symptoms. Furthermore, our study highlights the potential of targeting the DNA repair pathways as a therapeutic strategy in ALS. By restoring or enhancing DNA repair efficiency in motor neurons, it may be possible to reduce the rate of neuronal loss and slow the progression of the disease. This approach could offer a promising avenue for the development of ALS treatments, focusing on the underlying molecular mechanisms. The broader implications of these findings extend to other neurodegenerative diseases where similar mechanisms of DNA repair impairment may be at play, potentially opening up new research directions and therapeutic opportunities in the field of neurodegeneration. Although more studies are needed to validate its therapeutic potential, the targeted delivery of Lig1 to mitochondria represents a significant step towards addressing the unmet medical need not only in FUS-associated ALS/FTD, but also in other disorders associated with mitochondrial dysfunction.

## Methods

### Cell lines, cell culture, and tissue origin
Human embryonic kidney HEK293 (ATCC) cell lines were cultivated in Dulbecco's modified Eagle's media (DMEM), containing 10% fetal bovine serum and 100 U/ml penicillin, and 100 U/ml streptomycin. FUS KO HEK293 cell line was described before[13]. Human fibroblasts were grown in DMEM/F12 media with 10% fetal bovine serum, 1.6% Sodium Bicarbonate (Corning), and 1% MEM non-essential amino acids (Gibco TM)[31]. The origin and the conversion of the patient-derived FUS mutant fibroblasts as well as the mutation corrected isogenic lines into Human iPSC cells were described previously[31].

Spinal cord autopsy tissue specimens from ALS patients and age-matched controls were obtained from the Department of Veteran's Affairs (VA) Biorepository in the USA. The details of patient specimens are listed in Supplementary Table 1. These studies were conducted in accordance with the ethics board standards at the VA and the institutional review boards at the Houston Methodist Research Institute (Houston, Texas).

### Induction of NPSCs and differentiation of motor neurons from iPSC lines
Control and FUS P525L iPSCs were cultured in dishes coated with Geltrex LDEV-Free, basement membrane matrix, using 1X Essential 8 media (cat#A1517001). These cells were maintained at 37 °C in a 5% CO$_2$ atmosphere. To derive NPSCs, PSC neural induction media (Gibco A1647801) was used as per the manufacturer's protocol. The process involved replacing the Essential 8 media with PSC neural induction media approximately 24 hours after sub-platting the iPSCs. This media was maintained for 7 days. The first passage (P0) NPSCs were then transferred onto Geltrex (Thermo Fisher) coated 6-well plates and cultured in StemPRO neural stem cell SFM media (A1050901). Neural induction efficiency was assessed at third passage by immunofluorescence staining, with a neural lineage stem cell marker (Nestin) and a pluripotent marker (Oct4). Motor neurons were derived from iPSCs obtained from WiCell Research Institute and VIB-KU Leuven[31]. The differentiation process followed established protocols with some modifications. In brief, iPSC clones were transferred from a 60-cm dish to a T-25 flask filled with neuronal basic media. The media consisted of a mixture of 50% Neurobasal media and 50% DMEM/F12 media, supplemented with N2 and B27 supplements without vitamin A. Collagenase type IV digestion was performed to facilitate suspension of the iPSC clones. Afterward, the suspended cell spheres were subjected to a series of incubations. Initially, they were treated with various inhibitors including 5 μM ROCK Inhibitor (Y-27632), 40 μM TGF-β inhibitor (SB 431524), 0.2 μM bone morphogenetic protein inhibitor (LDN-193189), and 3 μM GSK-3 inhibitor (CHIR99021). This was followed by incubation in a neuronal basic media containing 0.1 μM retinoic acid (RA) and 500 nM Smoothened Agonist (SAG) for 4 days.

Subsequently, the cell spheres were incubated for 2 days in a neuronal basic media containing RA, SAG, 10 ng/ml Brain-derived neurotrophic factor (BDNF), and 10 ng/ml Glial cell-derived neurotrophic factor (GDNF). To dissociate the cell spheres into single cells, they were exposed to a neuronal basic media containing trypsin (0.025%)/DNase in a water bath at 37 °C for 20 min. Afterward, the cells were pipetted into a media containing ROCK inhibitor (1.2 mg/ml) to maintain their viability. Following cell counting, a specific number of cells were seeded onto dishes or chamber slides coated with 20 μg/ml Laminin. These cells were incubated for 5 days in a neuronal basic media containing RA, SAG, BDNF, GDNF, and 10 μM DAPT. Subsequently, the media was switched to one containing BDNF, GDNF, and 20 μM Inhibitor of γ-secretase (DAPT) for an additional 2 days. For motor neuron maturation, the cells were cultured in a media containing BDNF, GDNF, and 10 ng/ml ciliary neurotrophic factor (CNTF) for a period exceeding 7 days.

## Antibodies and plasmids

Rabbit anti-FUS (Cat# A300–302 A) and anti-Lig1 (A301-136A) antibodies were procured from Bethyl Laboratories, Inc. Mouse anti-FLAG antibody (A8592) was obtained from Sigma-Aldrich, and mouse anti-Lig3 antibody (Cat# ab587) was purchased from Abcam. Mouse anti-Tom20 (SC-17764), anti-HSP60 (SC-13115) and anti-PCNA (SC-56) antibodies were procured from Santa Cruz Biotechnology. Rabbit anti-CYTB (Cat# 55090-1-AP), COX1 (Cat# 13393-1-AP) and ND4 (Cat# 26736-1-AP) procured from Protein tech. Fluorescent secondary antibodies, Alexa Fluor 488 anti-mouse (Cat# A28175), and Texas Red anti-rabbit antibody (Cat# T-2767) were obtained from Life Technologies. The antibodies were diluted at 1:1000 for western blotting, 1:500 for immunofluorescence, and 1:100 for PLA.

The human Lig1 coding sequence was re-cloned into pCDNA3.1 plasmid as an N-terminal FLAG and COX8 gene mitochondrial targeting sequence containing construct (FLAG-MTS-Lig1)[56].

## Mouse models and genotyping

Mouse experiments were approved by the institutional ethics review board of Houston Methodist Research Institute. FUS WT (Strain#:017916) and R495X (Strain#:017928) transgenic mice were obtained from the Jackson Laboratory repository. Animals were propagated and genotyped following the guidelines provided by Jackson Laboratory. The mice were maintained under constant conditions (21 ± 1 °C; 60% humidity) with a 12/12-h light/dark cycle and given unrestricted access to food and water. Mice were weaned at 21 days and genotyped by ear biopsy.

## Transfection, immunoblotting, and immunofluorescence

Fibroblast cells were transfected with plasmids using Lipofectamine 3000 (Invitrogen), according to the manufacturer's instructions. NPSC cells were transfected with Lipofectamine™ Stem transfection reagent (Cat#STEM00001) (Thermo Fischer), according to manufacturer's instructions. Immunoblotting and immunofluorescence were performed as per standard protocols[57,58]. For immunoblotting, cell lysates extracted with 1X RIPA buffer (Millipore) containing the protease inhibitor cocktail (Roche) were loaded into 4–12% Bis-Tris precast (Bio-Rad) gels for electrophoresis. After transfer to nitrocellulose membranes, the separated proteins were incubated with primary and secondary antibodies, and the protein signals were detected by adding chemiluminescence reagents (LI-COR) and visualized by LI-COR Odyssey imaging system. For immunofluorescence, cells grown on chamber slides were first fixed with 4% paraformaldehyde for 15 min, followed by permeabilization in 0.5% Triton X-100 for 15 minutes. The cells were then incubated with primary antibodies overnight and with fluorescently labeled secondary antibodies for 2 h. Finally, the immunofluorescent images were captured by ZEISS Axio Observer fluorescence microscope.

## Co-immunoprecipitation (co-IP)

The co-IP of endogenous FUS was performed using protein A/G PLUS agarose beads (Santa Cruz Biotechnology). Cells were harvested and lysed with a buffer containing 0.2% NP-40, 150 mM NaCl, 25 mM Tris-HCl, and 0.1% SDS. The lysate was precleared by adding 1 μg of a control antibody IgG along with the protein A/G beads, which helps to remove non-specific binding. After a 30-minute incubation at 4 °C, the supernatant was collected and incubated with a primary antibody suitable for FUS. Next, the protein A/G beads were added to the mixture and incubated overnight in a rocker platform. The protein-antibody-bead complex was then centrifuged at 600 g to separate the beads from unbound proteins. The beads were washed three times to remove any remaining unbound proteins and the protein complex was eluted from the beads and subjected to IB analysis[13].

## In situ Proximity Ligation Assay (PLA)

An in-situ PLA assay was performed using a Duolink PLA kit (Sigma) following the manufacturer's instructions[59,60]. Briefly, cells grown in chamber slides were fixed with 4% formaldehyde for 15 min at 37 °C, permeabilized with 0.5% TritonX-100 for 10 min, and then incubated with primary antibodies overnight. The cells were then incubated at 37 °C with PLA probes for 1 h, with ligase for 30 min, and with polymerase for 100 min. The slides were mounted with mounting medium containing DAPI and the PLA signal was visualized using a fluorescence microscope (ZEISS Axio Observer) or a confocal microscope (Olympus, FV3000). The negative control involved incubation with IgG.

## Long amplicon PCR (LA-PCR)

Genomic DNA was extracted using Qiagen Blood and Tissue kit following the manufacturer's instructions[61]. In this study, two sets of primers with a range of 9 kb were used for both human and mouse samples covering the entire ~16 kb mtDNA. Additionally, a control amplification of 250 bp short segment was conducted. Post-amplification, the PCR products were analyzed using two independent methods: Agarose gel electrophoresis and the Quant-iT™ Pico-green DNA quantification assay (Thermo Fisher)[62]. The primer sequences are listed in Supplementary Table 2. The determination of mitochondrial copy number variations among samples was conducted as previously described[63]. Briefly, total genomic DNA extracted using Genomic-tip 20/G (QIAGEN) from respective samples was used as a template to set up quantitative real-time PCR (rtPCR) on Applied systems 7500 Real-Time PCR system. Three independent reactions with primer sets of mtDNA (ND1 and 16sRNA) along with genomic primers for nuclear genome quantification (Primers sequences in Supplementary Table 2). Mitochondrial DNA copy number was determined relative to nuclear DNA.

## In vitro ligation activity assay

DNA oligos were synthesized by Sigma using the following sequences: p24-Cy3-GGCACGGTCTACACGGCACACGAG, p27-TGTACATGATACGATTCCAAGCTAAGC, and p51-CCGTGCCAGATGTGCCGTGTGCTCACATGTACTATGCTAAGGTTCGATTCG. The assay was performed according to a previously published protocol[13]. Briefly, 10 pmol of each oligomer was incubated with 50 mM NaCl in a heated water bath until the boiling water cooled to room temperature. Annealed oligomers were mixed with various mitochondrial extracts in 1× T4 ligation buffer and the mixture was incubated in a water bath for 20 min at 30 °C. Samples were mixed with 2× TBE buffer, heated for 3 min at 100 °C, and cooled down on ice for 3 min. Oligomers were then separated by denaturing urea polyacrylamide gel electrophoresis. The band with Cy3 fluorescence was detected by Typhoon FLA 7000 fluorescence imaging system[64,65].

## Mitochondria protein extraction

Mitochondrial proteins were isolated using the differential centrifugation method as described by Ivan Dimauro et al. [66]. All

procedures were performed at 4 °C or on ice. The collected cells were homogenized in STM buffer (250 mM Sucrose, 50 mM Tris-HCl at pH 7.4, 5 mM MgCl$_2$, and a protease inhibitor) using a Glass-Col homogenizer set to 700–1000 rpm. The homogenate was collected in a centrifuge tube and kept on ice for 30 minutes followed by centrifugation at 800 g for 15 min. The supernatant was collected, and centrifugation was repeated to isolate mitochondrial and cytosolic fractions. The supernatant was then centrifuged at 11,000 g for 15 min to pellet mitochondria and supernatant containing cytosolic fraction. The pellet, washed and centrifuged (at 11,000 g for 10 min) with STM buffer, was then suspended in 1× RIPA buffer containing 50 mM Tris HCl pH 6.8, 1 mM EDTA, 0.5% Triton-x-100, and a protease inhibitor for lysis followed by centrifugation for 15 min at 15,700 g.

## ChIP assay

ChIP assay was performed according to a previously published protocol[64]. In brief, patient-derived fibroblasts with FUS WT and FUS P525L mutation were treated with 100 ng/ml glucose oxidase for 1 h. Mitochondria isolation was performed, as previously described, up to the pellet stage. The isolated mitochondria were crosslinked in 1% formaldehyde for 20 min at room temperature. The crosslinking reaction was quenched with 125 mM glycine, and the mitochondria were harvested using cold 1× PBS/1× protease inhibitor buffer. Chromatin was fragmented into average sizes of 250–650 bp by sonication. The Lig3 ChIP assay was performed using cleared lysates, where 5 µg of Lig3 antibody and the Magna ChIP Protein A magnetic beads were incubated overnight at 4 °C. All the ChIP eluates were subjected to reverse-crosslinking, purification by phenol/chloroform extraction, and were finally dissolved in 10 mM Tris-HCl (pH 8). The relative occupancy of the target protein at sites of mtDNA damage was analyzed by qPCR with primers (sequences given in Supplementary Table 2) targeting the mitochondrial ND1 and 16 s rRNA genes.

## Measurement of mitochondrial membrane potential (ΔΨm)

Patient-derived fibroblasts and NPSC cells were seeded in 96-well plates and exposed to 100 ng/ml GO for 1 h. After treatment, the cells were allowed to recover for 2 hours. The mitochondrial membrane potential was then measured using a TMRE mitochondrial membrane potential assay kit, according to the manufacturer's instructions[67].

## Cellular mitochondrial respiratory metabolic phenotype assessment

Mitochondrial respiration measurement in patient derived fibroblasts and NPSC cells were performed in untreated, 1 h GO treated, recovery followed by GO treatment, 30 minutes NaAs treated, recovery after NaAs treatment. 25,000 cells per well were plated and assessed by measurement of OCR in response to inhibitor and uncoupler injections following the Seahorse XF96 Cell Mito Stress Test protocol[68]. Briefly, OCR was measured three times at the baseline condition and ATP synthase inhibition was attained by injecting oligomycin. The decrease in OCR corresponds to oxygen consumption serving ATP production. Then, the cells were injected with mitochondrial uncoupler carbonilcyanide p-triflouromethohyphenylhydrazone (FCCP) to force cells to utilize their maximal respiration capacity to preserve mitochondrial membrame potential. The increased OCR reflects the reserve capacity of mitochondrial electron transport. The third injection consisted of adding Antimycin-A, a complex-III inhibitor to cause a shutdown in electron transport chain and to abolish mitochondrial oxygen consumption, enabling the calculation of proton leakage as a measure of difference in OCR after injection of oligomycin and antimycin-A.

## Mitochondrial DNA sequencing using REPLI-g followed by NGS

Genomic DNA was extracted from HEK293, patient derived fibroblasts, and human tissues using the Qiagen blood and tissue kit and following the manufacturer's protocol. The total DNA was then amplified with REPLI-g Mitochondrial DNA kit according to the manufacturer's recommendation (Qiagen, Germantown, MD)[69,70]. The resulting PCR products were sequenced on Illumina HiSeq platform by using the GENEWIZ next generation sequencing service. For the sequencing, DNA samples pooled from three different sets of REPLI-g DNA products were used for cell lines. For the human tissues, REPLI-g products from three controls or three patient samples were combined to achieve the maximum coverage of mutations. After the sequencing analysis, unique mitochondrial variations (including SNPs, insertions, and deletions) were identified by the in-house experts at GENEWIZ and the variant call files (VCF) were provided after the analysis. We further analyzed these VCF files confirm the sequence changes and elucidate the impact of specific variation. To determine their effect on protein encoding capabilities, each variation was categorized using an online tool called PolyPhen-2.

## Statistical analysis

At least three independent experiments were performed for each set of presented data. Statistical analysis was performed using GraphPad Prism software. The results were analyzed for significant differences using Student's t-tests. A $p$-value of less than 0.05 was considered statistically significant.

## Reporting summary

Further information on research design is available in the Nature Portfolio Reporting Summary linked to this article.

## Data availability

All unique and stable reagents produced in this research can be obtained from the corresponding author by completing a Materials Transfer Agreement. Any further information is needed to reanalyze the data presented in this paper, it can be requested from the corresponding author. The Mitochondrial DNA sequencing data have been submitted to the Genome variation map (https://ngdc.cncb.ac.cn/gvm/) and are accessible through GVM accession number GVM000579. The original imaging and raw data generated during this study are provided as Source Data. Source data are provided with this paper.

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

## Acknowledgements

This research is primarily supported by the National Institute of Neurological Disorders and Stroke (NINDS) and the National Institute of Aging (NIA) of the National Institutes of Health (NIH) under award number RF1NS112719 to M.L.H. Research in the Hegde laboratory was also supported by the NIH awards R01NS088645, R03AG064266, and R01NS094535, as well as the Sherman Foundation Parkinson's Disease Research Challenge Fund and Houston Methodist Research Institute's internal funds. M.L.H. acknowledges Everett E. and Randee K. Bernal for their support via Centenial Endowed Chair of the Neurological Institute. A.E.T. acknowledges the NIH award ES012512. L.V.D.B. acknowledges support from the KU Leuven (C1 and "Opening the Future" Fund), the ALS Liga (A Cure for ALS), and the Generet Award for Rare Diseases. W.G. acknowledges the support from IdEx Unistra (ANR-10-IDEX-0002) under the framework of the French Program 'Investments for the Future'. The authors would like to express their gratitude to members of Hegde laboratory for various assistance and Drs. Gillian Hamilton and Anna Dodson at Houston Methodist Research Institute (Houston, TX) for their assistance with document editing and their financial support to Walters educational initiative.

## Author contributions

M.K. performed most of the experiments with assistance from H.W., W.G., J.M., P.M.H., V.H.M.R., V.P., I.V., A.Z. Additionally, M.K. contributed to data analysis and interpretation and co-wrote the manuscript. W.G., S.M., A.E.T., D.J.H., and L.V.D.B. provided important reagents, intellectual insights, and manuscript feedback. M.L.H. designed and supervised the study, analyzed and interpreted the data, and cowrote and prepared the final manuscript. All authors participated in result discussions and provided feedback on the manuscript.

## Competing interests

The authors declare no competing interests.

## Additional information

[1]Division of DNA Repair Research within the Center for Neuroregeneration, Department of Neurosurgery, Houston Methodist Research Institute, Houston, TX, USA. [2]KU Leuven-Department of Neurosciences, Experimental Neurology and Leuven Brain Institute (LBI), Leuven, Belgium. [3]Stem Cell Institute, Department of Development and Regeneration, KU Leuven, Leuven, Belgium. [4]INSERM, UMR-S1118, Mécanismes Centraux et Périphériques de la Neuro-dégénérescence, Université de Strasbourg, CRBS, Strasbourg, France. [5]College of Medicine, Texas A&M University, College Station, TX, USA. [6]Center for Bioenergetics, Houston Methodist Research Institute, Houston, TX, USA. [7]Department of Medicine, Houston Methodist, Weill Cornell Medicine affiliate, Houston, TX, USA. [8]Departments of Internal Medicine, and Molecular Genetics and Microbiology and University of New Mexico Comprehensive Cancer Center, University of New Mexico, Albuquerque, NM, USA. [9]VIB, Center for Brain & Disease Research, Laboratory of Neurobiology, Leuven, Belgium. [10]Department of Neuroscience, Weill Cornell Medical College, New York, NY, USA. [11]These authors contributed equally: Manohar Kodavati, Haibo Wang. ✉e-mail: mlhegde@houstonmethodist.org

