## [Peer Review File · Nature Communications]

FUS Unveiled in Mitochondrial DNA Repair and Targeted Ligase-1 Expression Rescues Repair-Defects in FUS-Linked Motor Neuron DiseaseReviewer #1 (Remarks to the Author):

The manuscript by Kodavati and colleagues demonstrates how disease-causing mutations in FUS impact DNA repair efficiency in amyotrophic lateral sclerosis (ALS). ALS is a devastating motor neuron disease which is poorly understood. Pathogenic mutations in FUS, an RNA binding protein, has been linked with a juvenile onset and aggressive form of ALS. Current study centers on the role of FUS, a crucial RNA-binding protein associated with ALS pathogenesis, and its potential impact on DNA repair efficiency. The authors found that pathogenic mutations in FUS severely compromised mitochondrial DNA repair efficiency in iPSC derived neurons. Additionally, the study sheds light on the functional interaction between FUS and mt Lig3, providing mechanistic guidance for potential therapeutic interventions using targeted expression of DNA Ligase 1 in mitochondria. This research holds promise for unraveling the molecular intricacies underlying ALS and opens up new avenues for targeted treatments. The importance of FUS in mitochondrial dysfunction can also have implications in other pathologies associated with FUS, including cancers. While the study presents substantial contributions, I have the following concerns, mostly minor, that the authors should address to further strengthen the conclusions and impact of this work.

1. While several experiments are performed in ALS-relevant patient-derived fibroblast cells, only key data are shown in motor neurons. The authors should validate the key findings shown in fibroblasts in ALS-relevant motor neurons to enhance the clinical relevance and overall quality of the paper.
2. Supplementary Figure 1c (FUS localization between WT and mutant motor neurons) should be included in the main figure along with immunoblotting using motor neurons.
3. The authors should perform the LA-PCR in figures 3b and 3f in motor neurons in untreated and GO-treated conditions.
4. PLA between FUS and Lig3 should be supported by Co-IP in patient-derived cells.
5. Figure 5b and Supplementary Figure 5a involving isogenic lines should also be performed in motor neurons.
6. Considering the observed compromised DNA repair efficiency in the mutant mice, it would be intriguing to investigate FUS protein levels in mitochondria of normal mice, FUS WT, and FUS R495X mice models.
7. It would be valuable to discuss the potential clinical implications of the study's findings. How might the compromised DNA repair efficiency observed in ALS-relevant cells impact disease progression and neurodegeneration?

Reviewer #2 (Remarks to the Author):

The study here aims to investigate the role of endogenous wildtype FUS in mitochondrial DNA (mtDNA) repair and elucidate the mechanisms by which FUS mutations contribute to mtDNA damage and neurodegeneration. Evidence is presented showing localization of endogenous FUS to mitochondria, with mutant FUS forms potentially collecting in mitochondria to a greater extent. Following oxidative stress, mutant FUS forms can be more prone to aggregation, reducing mitochondrial localization and causing mitochondrial dysfunction. Additionally, reduced mtDNA integrity and increased mutations were found in various FUS mutant material, as was impaired mtDNA repair. Mutant FUS proteins had reduced interactions with Lig3, coinciding with reduced total mitochondrial ligation activity and lower association of Lig3 with mtDNA in FUS mutant cells, providing a underlying molecular mechanism for the reported mitochondrial defects. A mitochondrial-tagged Lig1 expression system is shown to correct the mitochondrial ligation defects, suggesting a possible approach to treating ALS.

While the studies are generally soundly conducted, they largely build off prior knowledge and represent a limited advance. For instance, FUS is known to localization to mitochondria, with abnormal localization patterns often seen for mutant FUS forms, and mitochondrial dysfunction has been reported in FUS mutant cell models. The novelty is largely in the observation of mtDNA repair defects,

yet that would likely be expected since FUS interacts with Lig3 (previous work by main author), a protein found both in the nucleus and mitochondria. Lastly, the observation that overexpression of a mitochondrial-targeted DNA ligase rescues the mtDNA repair defects of FUS mutant cells is not surprising. I have the following specific comments as well:

1. Figure 1. In panel A, why is PCNA found in the cytosol? Is it not a nuclear marker? In B, GO is argued to promote mitochondrial localization of WT FUS; however, that's not apparent in the graph. Is there an explanation? In C, it would be nice for the cells themselves to be visible (not just DAPI), to verify that the PLA signal is indeed in the cytosol. Mitotracker staining might also be useful to show colocalization of PLA signals and mitochondria.
2. Figure 3. MT LA and MT SA should be explicitly defined in the legend. Based on the data shown, reduce mtDNA integrity seems to be most pronounced in the FUS mutant expressing cells relative to the clean KO cells. Do the authors have an explanation? Could there be dominant negative effects of the mutant proteins? Also, is the nature of the FUS mutation known in the ALS patient samples? Can that be reported somewhere in the text and is there anything informative there? Finally, it would be more informative to show the exact nature of the mutations [nucleotide substitutions (e.g., A to T), deletions, etc.] as opposed to "possibly damaging", "silent", etc. Please consider including full details in the report.
3. Figure 4. Assuming I understand the assay correctly, I would be careful about interpreting the reduced chromatin association of Lig3 as reduced localization to DNA damage sites. Although likely, it's technically unknown whether DNA damage is actually present in the IPed DNA material. Additionally, assuming DNA damage is present, would this not reduce PCR efficiency and thus interfere with the final results? Some careful thought about the presentation of these data is recommended.
4. Figure 5. In panel A, again, as noted above, co-images with mitotracker might be more informative than DAPI. In panel B, some quantitation would be useful, as it appears that there is a slightly higher level of P525L in the mitochondrial fraction relative to the other two proteins.
5. The authors argue that gene editing approaches to treat ALS are "technically difficult", yet they imply that treating patients with a mitochondrial-targeted Lig1 expression system is more practical. Moreover, what is the clinical evidence for "the high specificity and low toxicity of Lig1-based therapy make it a safe and effective treatment option."? I think these parts need to be more carefully presented or at least justified.
6. It's unclear to me how FUS would mediate recruitment of Lig3 to DNA. Is FUS a damage-specific DNA binding protein? Does it have preference for SSBs? Maybe some of the molecular mechanism could be more clearly described, as a general DNA binding property does not obviously make FUS a repair scaffold. For example, XRCC1, the benchmark scaffold, has been reported to have increased specificity for DNA strand breaks.
7. There are numerous text problems, including typos, errors, poor organization and poor grammar. I highlight just a few issues below, but the manuscript requires rigorous proofreading and editing.
 - a. Results, first section. Makers should be markers. Also, the sentence beginning "Although we did not observe significant differences..." is not a sentence; please rework. In the following sentence, "However, ...", which cells are the authors referring to?
 - b. Results, mtDNA repair section. "role of FUS role" needs to be edited. In the same sentence, SSB is defined a second time and "activity of break-sealing DNA Lig3" is awkward; please correct.
 - c. Discussion. Lower DNA repair does not make mtDNA more susceptible to DNA damage, it just means DNA damage is removed with less efficiency and thus more persistent.
 - d. Figure 6. Green "oval". ntDNA should be mtDNA.

Reviewer #3 (Remarks to the Author):

Summary:

Kodavati and colleagues aim to elucidate the role of FUS in mediating mitochondrial DNA damage repair and highlight implications in the development of ALS. Using various cell lines, mouse models, and ALS patient samples, the authors describe that endogenous FUS interacts with mitochondrial localized Lig3 and recruits Lig3 to sites of mtDNA damage to effect DNA repair, particularly in settings of oxidative damage. They also describe that ALS-related FUS mutants impair mtLig3 activity, leading to increased mtDNA damage, which can be restored by overexpression of a mitochondria-targeted Lig1. The authors conclude FUS plays a critical role in mtDNA repair and implicate mtDNA damage in the development of FUS-associated neurodegeneration.

Overall Comments:

The authors build upon their previous work implicating a role for FUS and Lig3 interactions in nuclear DNA repair. Here, they extend their results to show that FUS now binds to the mt-localized isoform of Lig3, this interaction appears to be impacted by ALS-related FUS mutations, and the FUS-mtLig3 complex is distinct from partners found at FUS-Lig3 sites within the nucleus. Further, mtLig3 is less abundant following FUS deletion. Overall, this study provides novel and intriguing insights into the role of FUS in maintaining mitochondrial genome integrity in neurodegeneration. There are notable areas of improvement for this study detailed below that do not detract from enthusiasm for the work and that if corrected, would strongly improve the rigor and description of the data herein, as well as some minor comments to improve the clarity of the work.

Major Comments:

1. The authors describe reduced mtDNA/nuclear DNA ratios for the P525L FUS mutation in Figure S3. Reductions in mtDNA would seem the most important consequence of impaired mtDNA repair that would ultimately explain reduced respiration, yet these data are not more prominently featured. Further, mtDNA/nDNA ratios are not reduced in FUS knockouts. It would be helpful if the authors showed these data in a more useful format (the averaged mean WT and mutants in all conditions side by side on the same graph, including in Figures S3D and S3E).
2. As a corollary to the above, mtDNA damage should also lead to reduced mtRNA expression and mitochondrial encoded proteins, beyond reduced mtDNA content. Again, these should be measured and shown to better explain the functional respiratory defects found in FUS mutants.
3. There currently is a heavy reliance on 293T mutant FUS cells and iPS-derived fibroblasts for many endpoints in the study which limit the impact of the work, as ALS manifests in neurons. 293T cells and fibroblasts are notorious for being far less dependent on mitochondrial function for cell health compared to neuronal cells. The authors should perform confirmatory studies of their work (such as functional studies in Figure 2, Lig3 activity assays in Figure 4, and rescue studies with Lig1 in Figure 5) in neuronal cell types.
4. Overall, Figure 3 was a challenge to review. Layouts of all figures were varied throughout the panel and tough to follow (F appears above E, for instance). It would help if this figure was laid out more clearly or split into more figures.
 - a. The text describing Figure 3E states "The results showed that mtDNA integrity increased significantly at 180 minutes in FUS WT cells, indicating successful DNA damage repair, while repair capacity was reduced in FUS KO cells (Fig. 3e)". This interpretation is misleading. The integrity appears to recover in both groups, while KOs appear worse because of a stronger decline in integrity at Mt LA at 0.5 h. This would suggest the KO is more susceptible to damage rather than unable to repair normally.
 - b. Further, it would help if the samples in 3E and F were directly compared (Y axes between these studies and graphs are shown in different formats) as it appears that the FUS ALS mutants have worsened recovery than the FUS mutants. Could this suggest a dominant negative effect of these mutants?
 - c. Figure 3G is a very confusing experiment as it is unclear what is the control group (is "patient tissue

the control?" and the layout and Y-axes are very different for each tissue type. There are also no error bars shown. How many times were these experiments performed?

5. For the FUS proximity ligation assays, several missing controls detract from the results. FUS PLA should be performed on FUS KO cells as a true negative control to ensure non-specific signals are not observed in the absence of one of the interacting proteins. A cytoplasmic or membrane counterstain/marker should also be included to ensure that PLA signal truly are intracellular and do not arise non-specifically elsewhere on the slide. Additionally, the Methods section does not indicate if Z stack images were captured. Imaging Z-stacks would ensure a more accurate representation of PLA signal in 3 dimensions on the imaged cells.

6. Population sizes used in each group and the specific statistical tests applied should appear in the figure legend for ease of the reader. Additionally, please include the specific data points that comprise the mean values in the bar graphs (as well as both upper and lower error bars). The appearance of the current data is rather opaque as currently shown.

7. In Figure 5C, data with use of the mt targeted Lig1 construct is confusing. Following cleavage of the MTS (and the Flag tag), Flag would be expected to diffuse out of the mitochondria. However, Flag is visualized in the mitochondria on the cell fractionation experiment without molecular mass markers (these are missing in all blots in the paper and ideally should be included). Is this full length mt Lig1?

Minor Comments:

1. The last part of the title is somewhat misleading. This study does not look at neurodegeneration, but rather the mutations associated with it. Please alter the title to accurately reflect the findings.
2. In most IF images throughout the paper, the scale bar in the bottom right corner appears to be placed on top of another scale bar.
3. Please provide quantitation of Figure 1A showing increase in FUS levels in mitochondria after GO stimulation.
4. The text in the Results describing Figure 1B should mention that there seems to be more P525L FUS in the mitochondria at baseline.
5. Figure 2 is overcrowded and would be easier to read if bar graphs and font were slightly smaller. Group labels are only shown at the bottom of Figure 2E which also was confusing to read.
6. The text states "Given the localization of FUS in mitochondria and the colocalization of mutant FUS in SGs induced by sodium arsenite, we hypothesized that the accumulation of mutant FUS in SGs results in reduced FUS levels in mitochondria, thereby disturbing its normal function". While the Seahorse did assess function, it did not assess whether FUS levels are reduced in the mitochondria after NaAs treatment.
7. The Figure 2 legend indicates that "Non-mitochondrial oxygen consumption was determined by measuring difference between total oxygen consumption and oligomycin induced reduction in oxygen consumption". Non-mitochondrial oxygen consumption is generally accepted as the oxygen consumption remaining after (ideally both) Rotenone and Antimycin A are added.
8. The Figure 2 legend indicates that "maximal respiration was assessed following mitochondria uncoupling by FCCP". Maximal respiratory capacity is generally measured as the difference between oxygen consumption following FCCP and oxygen consumption following Rotenone and Antimycin A. Please clarify.
9. Figure 3 legend states "e shows patient derived fibroblasts whereas f represents HEK 293 cells." Should this not read that e represents the HEKs and f represents the fibroblasts?
10. In Figures 4A and B, please provide a quantification of Lig3 levels in each of these groups as well as for FUS levels in Figure 5B.
11. In the text "We first tested the overall ligation activity in mitochondria by in vitro ligation activity assay and found that MTS-Lig1expressing fibroblasts with FUS P525L mutation had a significantly higher ligation activity than the control (Fig. 6d)". This appears to be mislabeled instead of Figure 5D.

Our Responses to the Reviewers' Comments

Reviewer #1:

General Comment: The manuscript by Kodavati and colleagues demonstrates how disease-causing mutations in FUS impact DNA repair efficiency in amyotrophic laterals sclerosis (ALS). ALS is a devastating motor neuron disease which is poorly understood. Pathogenic mutations in FUS, an RNA binding protein, has been linked with a juvenile onset and aggressive form of ALS. Current study centers on the role of FUS, a crucial RNA-binding protein associated with ALS pathogenesis, and its potential impact on DNA repair efficiency. The authors found that pathogenic mutations in FUS severely compromised mitochondrial DNA repair efficiency in iPSC derived neurons. Additionally, the study sheds light on the functional interaction between FUS and mt Lig3, providing mechanistic guidance for potential therapeutic interventions using targeted expression of DNA Ligase 1 in mitochondria. This research holds promise for unraveling the molecular intricacies underlying ALS and opens up new avenues for targeted treatments. The importance of FUS in mitochondrial dysfunction can also have implications in other pathologies associated with FUS, including cancers. While the study presents substantial contributions, I have the following concerns, mostly minor, that the authors should address to further strengthen the conclusions and impact of this work.

Response: Thank you for your insightful comments and for recognizing the potential impact of our work. We have carefully reviewed and addressed each of your concerns, making appropriate revisions to the manuscript. Your constructive feedback and suggestions have significantly helped to improve the manuscript's clarity, depth, and overall impact. We appreciate the opportunity to address your concerns.

Comment-1: While several experiments are performed in ALS-relevant patient-derived fibroblast cells, only key data are shown in motor neurons. The authors should validate the key findings shown in fibroblasts in ALS-relevant motor neurons to enhance the clinical relevance and overall quality of the paper.

Response: Thank you for the valuable suggestion. We agree that validating key findings in ALS-relevant motor neurons is crucial for enhancing the clinical relevance and overall quality of the manuscript. Initially, due to the practical challenges associated with scaling up neuronal cultures, we opted to perform experiments necessitating a substantial volume of cells in fibroblasts, reserving procedures such as immunofluorescence or LA-PCR for motor neurons. In response your suggestion, we have now included additional experiments using ALS-relevant motor neurons, thus complementing our findings obtained from patient-derived fibroblast cells. These new data include PLA analyses (Fig. 1e) and immunoblotting conducted from motor neuron extracts (Fig. 1b). These new data in motor neurons are consistent with our data in fibroblast cells and help to reinforce the significance and applicability of our results in the context of ALS.

Comment-2: Supplementary Figure 1c (FUS localization between WT and mutant motor neurons) should be included in the main figure along with immunoblotting using motor neurons.

Response: Thank you for your suggestion regarding the placement of Supplementary Fig. 1c. We agree that highlighting the differences in FUS localization between WT and mutant motor neurons is crucial for a comprehensive understanding of our findings, and thus have performed PLA in motor neurons and counterstained it with MitoTracker (Fig. 1e).

Additionally, we have included immunoblotting data using motor neurons (Fig. 1b), further substantiating our findings.

Comment-3: The authors should perform the LA-PCR in figures 3b and 3f in motor neurons in untreated and GO-treated conditions.

Response: Thank you for your recommendation to perform LA-PCR in motor neurons under untreated and GO-treated conditions for Fig. 3b and 3f. In response to your suggestion, we have conducted the LA-PCR experiments in motor neurons under both untreated and GO-treated conditions. We have included these new data in the revised manuscript.

Comment-4: PLA between FUS and Lig3 should be supported by Co-IP in patient-derived cells.

Response: We agree with your feedback and have now included Co-IP data involving FUS and Lig3 in the Fig. 4d in NPSC cells.

Comment-5: Figure 5b and Supplementary Figure 5a involving isogenic lines should also be performed in motor neurons.

Response: Thank you for your constructive comment. We have now performed the IB experiments presented in Fig. 5b using motor neurons, LA-PCR presented in Fig. 5e and Supplementary Fig. 6a using both mutant motor neurons and their isogenic control lines.

Comment-6: Considering the observed compromised DNA repair efficiency in the mutant mice, it would be intriguing to investigate FUS protein levels in mitochondria of normal mice, FUS WT, and FUS R495X mice models.

Response: Thank you for your suggestion. We have included immunoblotting data from control, WT, and mutant FUS mice brain tissue (cortex) mitochondrial extracts, probed with a FUS antibody. Appropriate markers for cellular fractions were probed as loading controls (Supplementary Fig. 4f). These data show a comparable level of FUS in WT and mutant mice, which is approximately 2-fold higher than normal control mice, as expected. This supports our conclusion that the elevated mutant FUS in mitochondria causes Lig3 defects by reducing its DNA damage recruitment.

Comment-7: It would be valuable to discuss the potential clinical implications of the study's findings. How might the compromised DNA repair efficiency observed in ALS-relevant cells impact disease progression and neurodegeneration?

Response: We completely agree that discussing the potential clinical implications of our findings is crucial for contextualizing our study in ALS to the broader field of neurodegeneration. In response, we have expanded the discussion section to include a more detailed account of how the observed compromised mitochondrial DNA repair efficiency in ALS-motor neurons might impact or contribute to overall disease progression and neurodegeneration. Specifically, we delve into the possibility that the disrupted mitochondrial DNA repair machinery and resulting overall mitochondrial dysfunction could accelerate neuronal vulnerability and loss, thereby exacerbating ALS progression. Moreover, we discuss how our findings could pave the way for novel DNA repair-targeted therapeutic strategies aimed at restoring DNA repair efficiency in ALS patients, potentially offering new avenues

to slow disease progression and mitigate neurodegenerative symptoms.

We appreciate your feedback, as it has allowed us to strengthen the manuscript by including additional experiments in motor neurons along with additional controls and emphasizing the translational potential of our findings. We sincerely hope that our responses and the revisions made to the manuscript have effectively addressed your concerns.

Reviewer #2

General Comment: The study here aims to investigate the role of endogenous wildtype FUS in mitochondrial DNA (mtDNA) repair and elucidate the mechanisms by which FUS mutations contribute to mtDNA damage and neurodegeneration. Evidence is presented showing localization of endogenous FUS to mitochondria, with mutant FUS forms potentially collecting in mitochondria to a greater extent. Following oxidative stress, mutant FUS forms can be more prone to aggregation, reducing mitochondrial localization and causing mitochondrial dysfunction. Additionally, reduced mtDNA integrity and increased mutations were found in various FUS mutant material, as was impaired mtDNA repair. Mutant FUS proteins had reduced interactions with Lig3, coinciding with reduced total mitochondrial ligation activity and lower association of Lig3 with mtDNA in FUS mutant cells, providing a underlying molecular mechanism for the reported mitochondrial defects. A mitochondrial-tagged Lig1 expression system is shown to correct the mitochondrial ligation defects, suggesting a possible approach to treating ALS.

While the studies are generally soundly conducted, they largely build off prior knowledge and represent a limited advance. For instance, FUS is known to localization to mitochondria, with abnormal localization patterns often seen for mutant FUS forms, and mitochondrial dysfunction has been reported in FUS mutant cell models. The novelty is largely in the observation of mtDNA repair defects, yet that would likely be expected since FUS interacts with Lig3 (previous work by main author), a protein found both in the nucleus and mitochondria. Lastly, the observation that overexpression of a mitochondrial-targeted DNA ligase rescues the mtDNA repair defects of FUS mutant cells is not surprising. I have the following specific comments as well:

Response: Thank you for your thoughtful comments. We value your feedback, which we believe will enhance the quality and impact of our work.

We acknowledge your concerns regarding the novelty of our study. However, we would like to emphasize a few key points that highlight the unique contributions of our research:

1. **Exploration of FUS in Mitochondrial Genome Repair:** Our study is the first to explore the involvement of FUS in mitochondrial genome repair and maintenance. Previous works, including our own, have demonstrated the role of FUS in DNA single-strand break repair in the nucleus, but its role in mitochondria has not been elucidated before.
2. **Utilizing Multiple, Disease-Relevant Models:** The comprehensive use of multiple disease-relevant models, including CRISPR/Cas9 mediated FUS knockout (KO) HEK293 cells, ALS patient-derived cell lines, and human FUS R495X transgenic mouse models, have allowed us to robustly demonstrate the impact of FUS mutations in mtDNA damage and repair defects.
3. **Targeted Expression of DNA Ligase (Lig1) for Therapeutic Approaches:** Our work introduces a novel therapeutic approach using targeted expression of Lig1 in

mitochondria. This has been shown to restore mtDNA integrity and function, presenting a potential significant advancement in treating FUS-associated neurodegeneration and other related disorders. While it can be expected that in the context of ligation defect during FUS-associated neurodegeneration, as mutant FUS obstructs and prevents Ligase 3 functions, our studies suggest that mitochondria-targeted expression of an alternative DNA ligase (Lig1), which normally does not express in postmitotic cells and acts independent of FUS, is able to complement the lost ligation activity. This is an important finding and unveils an opportunity for DNA repair targeted intervention to restore mitochondrial functions in FUS-associated neurodegeneration.

4. **Broad Implications Beyond FUS-ALS:** The findings from our study could have broader implications, linking mitochondrial DNA damage and repair defects to a spectrum of neurodegenerative diseases, thereby contributing significantly to the field.

While we understand that some elements, like the localization of FUS to mitochondria, build on existing knowledge, we believe that our study offers substantial novel insights, especially in understanding the molecular mechanisms underpinning mitochondrial dysfunction in neurodegenerative diseases, and in proposing a new therapeutic strategy.

We hope that these clarifications address your concerns, and we are open to further discussions and revisions to improve our manuscript. Please find below detailed responses to each of your specific comments, illustrating how we have addressed them in the revised manuscript.

Comment-1: Figure 1. In panel A, why is PCNA found in the cytosol? Is it not a nuclear marker? In B, GO is argued to promote mitochondrial localization of WT FUS; however, that's not apparent in the graph. Is there an explanation? In C, it would be nice for the cells themselves to be visible (not just DAPI), to verify that the PLA signal is indeed in the cytosol. Mitotracker staining might also be useful to show colocalization of PLA signals and mitochondria.

Response: Thank you for your valuable feedback regarding Fig. 1. Please find below our responses to each specific point:

- **Regarding PCNA in panel A:** We agree that PCNA is traditionally recognized as a nuclear marker (by its nomenclature). However, several previous studies have reported the distribution of PCNA in both cytosolic and nuclear fractions (D. Bouayad et al., JBC,2012; S. Wisnovsky, et al., Nature chemical biology., 2016). In our study, PCNA was used as a control to ensure the absence of extramitochondrial contamination, particularly nuclear contamination, in our mitochondrial preparations. PCNA is a relatively abundant protein with a diffuse staining pattern in both the nucleus and cytoplasm. The established immunochemistry protocols usually used to detect nuclear PCNA extract the diffusible fraction leaving the chromatin bound PCNA fraction
- **Regarding panel B:** Thank you for your helpful suggestion. The WT FUS level in the mitochondria does indeed increase after GO treatment, as evident in the top western panel. The graph under panel C represents data from a FUS co-IP experiment. To address potential confusion, we have now included a quantitative histogram directly corresponding to the immunoblot in panel B. This addition illustrates the enhanced mitochondrial FUS levels in GO-treated cells.

- **Regarding the suggestion of MitoTracker staining:** We appreciate your suggestion. To enhance the visual validation, we have now included PLA along with MitoTracker staining in several experiments, including in Fig. 1d. We have repeated this experiment in mutant iPSC derived motor neurons and added data in Fig. 1e along with original fibroblast cell data with the new IF images. This addition helps to vividly display the colocalization of PLA signals and mitochondria, ensuring a more robust representation of our data.

Comment-2: Figure 3. MT LA and MT SA should be explicitly defined in the legend. Based on the data shown, reduce mtDNA integrity seems to be most pronounced in the FUS mutant expressing cells relative to the clean KO cells. Do the authors have an explanation? Could there be dominant negative effects of the mutant proteins? Also, is the nature of the FUS mutation known in the ALS patient samples? Can that be reported somewhere in the text and is there anything informative there? Finally, it would be more informative to show the exact nature of the mutations [nucleotide substitutions (e.g., A to T), deletions, etc.] as opposed to “possibly damaging”, “silent”, etc. Please consider including full details in the report.

Response: Thank you again for your keen observations and valuable suggestions. We have thoroughly revised our manuscript and figures to improve clarity and comprehensiveness based on your feedback.

Firstly, we have explicitly defined "MT LA" and "MT SA" in the legend as suggested.

Concerning the comparison between mutants and KO lines, we recognize the difficulty in making quantitative comparisons due to the separate gel analysis of LAPCR for mutants and KO lines. However, based on our mutation level data derived from sequencing, we did not observe a significant difference between KO and mutants in terms of mtDNA instability. We speculate that any slight increase in mtDNA instability in mutant cells could be attributed to enhanced overall oxidative stress and subsequent DNA damage caused by mutant pathology. A similar phenomenon involving enhanced nuclear genome instability associated with a TDP-43 mutant has been previously observed and discussed in our earlier work (Guerrero et al 2020).

While the ALS patient samples exhibit positive FUS pathology, they originate from patients without a familial history of ALS, and as such, specific mutation information is unfortunately not available.

We appreciate your suggestion regarding the sequencing data and have included specific details about the nature of mutations in Supplementary Table 3. This addition will undoubtedly offer a better understanding of the mutational landscape.

Comment-3: Figure 4. Assuming I understand the assay correctly, I would be careful about interpreting the reduced chromatin association of Lig3 as reduced localization to DNA damage sites. Although likely, it's technically unknown whether DNA damage is actually present in the IPed DNA material. Additionally, assuming DNA damage is present, would this not reduce PCR efficiency and thus interfere with the final results? Some careful thought about the presentation of these data is recommended.

Response: We appreciate your insightful comments and the opportunity to clarify the aspects related to our assay and its interpretation.

You are correct in your understanding of the assay. Regarding your point about DNA damage potentially reducing PCR efficiency, thus affecting the final results, this is a valid

observation. However, in our LA-PCR assay (Figure 3), the basal level damage, particularly in a short amplicon (SA, used as a control for the LA-PCR assay), appears minimal when contextualized within numerous copies of mitochondrial DNA obtained from a substantial number of cells. Such minimal damage does not manifest noticeably when amplifying a short amplicon (200-300 bp). This distinction becomes pronounced when larger segments (10 kb) are amplified, allowing for measurable DNA damage detection, a standard approach commonly applied in DNA damage measurements by LA-PCR (Claudia P Gonzalez-Hunt, et al., 2016; Bennett Van Houten, et al., 2006).

Similarly, taking a cue from the above fact, for the ChIP procedure, DNA is sonicated to achieve a fragmentation size of around 200-300 bp. Subsequent amplification of these short templates utilizing randomized primers minimizes the influence of DNA damage on the assay results, particularly under basal states. However, in acknowledgment of your comment, we conducted a control experiment involving the pulldown of Lig3 from mitochondrial extracts of both control and GO-treated cells, followed by similar ChIP analyses using identical primers as those applied in Fig. 4f. Our results indicate an enhancement of Lig3 enrichment at mitochondrial DNA upon GO treatment, suggesting that under our experimental conditions and assay design, DNA damage does not substantially impede the assay results. We have included these additional data in the Supplementary Fig. 5c and have refined the discussion and interpretation of these results in the manuscript to accommodate this consideration. We sincerely appreciate your observation, as it has significantly contributed to enhancing the robustness and clarity of our data interpretation.

Comment-4: Figure 5. In panel A, again, as noted above, co-images with mitotracker might be more informative than DAPI. In panel B, some quantitation would be useful, as it appears that there is a slightly higher level of P525L in the mitochondrial fraction relative to the other two proteins.

Response: Thank you. As already above in response to comment-1, we agree with your suggestion and have incorporated co-images with MitoTracker, in addition to DAPI in Fig. 1e.

We acknowledge your observation concerning the quantitation in Panel B, and we have included quantitative assessments to improve data interpretation. Your observation regarding the slightly elevated levels of P525L in the mitochondrial fraction compared to other proteins has been carefully considered and appropriately addressed in our revised representation.

Comment-5: The authors argue that gene editing approaches to treat ALS are “technically difficult”, yet they imply that treating patients with a mitochondrial-targeted Lig1 expression system is more practical. Moreover, what is the clinical evidence for “the high specificity and low toxicity of Lig1-based therapy make it a safe and effective treatment option.”? I think these parts need to be more carefully presented or at least justified.

Response: Thank you for your meticulous feedback, which has guided us toward a more nuanced and accurate representation of potential therapeutic approaches and their potential practicalities. We wish to clarify that our intention to convey the adaptability of Lig1 expression to an mRNA-based therapeutic approach. This flexibility appears promising, especially when considering potential challenges associated with gene therapy. Such a method could offer a versatile avenue to navigate around specific hurdles intrinsic to gene-editing tactics.

Regarding the safety and effectiveness of the Lig1-based approach: Your comment regarding the safety and effectiveness of the Lig1-based approach has prompted a reconsideration of how we present the speculative aspects of Lig1's safety and efficacy. We aimed to communicate that the vulnerability of DNA ligase activity during ageing, in conjunction with the essential role of ligase in DNA break sealing, could make Lig1 a potentially low-toxicity intervention in the context of older patients exhibiting diminished ligase activities. However, recognizing the absence of robust clinical validations at this stage, and to prevent any unintended overstatement of the current evidence supporting Lig1's therapeutic applicability, we have opted for a more cautious presentation, omitting portions that might imply a higher degree of clinical certainty and softening the language.

We are grateful for your insightful comments, which have been instrumental in refining our discussion for precision and factual alignment with the current stage of research and validation.

Comment-6: It's unclear to me how FUS would mediate recruitment of Lig3 to DNA. Is FUS a damage-specific DNA binding protein? Does it have preference for SSBs? Maybe some of the molecular mechanism could be more clearly described, as a general DNA binding property does not obviously make FUS a repair scaffold. For example, XRCC1, the benchmark scaffold, has been reported to have increased specificity for DNA strand breaks.

Response: Thank you for your insightful question, which provides us with an opportunity to further clarify the molecular mechanisms behind FUS's involvement in recruiting Lig3 to DNA). Building on existing knowledge from previous research (Haibo Wang, et al., 2018, Maria Vladislavovna Sukhanova, et al., 2020) including our own work, we have gained substantial insights into the role of FUS in single-strand break (SSB) repair within nuclear genomes. FUS is instrumental in facilitating the PARP-dependent recruitment of the XRCC1/Lig3 complex to SSB sites. Consequently, mutations in or the absence of FUS substantially impairs the recruitment and activity of Lig3. In our prior findings, we observed that mutant FUS exhibits a diminished capacity to localize to laser-induced damage sites within the nuclear genome. FUS has also been demonstrated to interact directly with Lig3, both in vitro and in vivo. Extending these observations to the mitochondrial genome, we speculate that the interaction between FUS and Lig3 is crucial for the efficient recruitment of Lig3 to damaged sites, an idea reinforced by our PLA and ChIP analyses.

It is imperative to highlight that in the mitochondrial context, where XRCC1 is absent, FUS assumes a significant role, likely analogous to the scaffolding function of XRCC1 in the nucleus, stabilizing the Lig3 complex. In light of your valuable feedback, we have refined our manuscript to better articulate the mechanistic involvement of FUS and its interaction with Lig3 in the DNA repair process. This enhanced clarity will foster a deeper understanding of the role of FUS in orchestrating DNA repair mechanisms. It is also crucial to acknowledge the inherent technical limitations in our ability to dissect mitochondrial repair mechanisms with the same depth as in nuclear genomes. This necessitates borrowing from established mechanistic insights related to nuclear Lig3 functions, which we have utilized as a framework for interpretation and understanding in our study.

Comment-7: There are numerous text problems, including typos, errors, poor organization and poor grammar. I highlight just a few issues below, but the manuscript requires rigorous proofreading and editing.

- a. Results, first section. Makers should be markers. Also, the sentence beginning “Although we did not observe significant differences...” is not a sentence; please rework. In the following sentence, “However, ...”, which cells are the authors referring to?
- b. Results, mtDNA repair section. “role of FUS role” needs to be edited. In the same sentence, SSB is defined a second time and “activity of break-sealing DNA Lig3” is awkward; please correct.
- c. Discussion. Lower DNA repair does not make mtDNA more susceptible to DNA damage, it just means DNA damage is removed with less efficiency and thus more persistent.
- d. Figure 6. Green “oval”. ntDNA should be mtDNA.

Response: Thank you for pointing out the text-related issues in the manuscript. We sincerely appreciate your attention to detail.

We have conducted a thorough review to address the typos, grammatical errors, and organizational inconsistencies you highlighted. Specifically:

- a. The term "makers" has been corrected to "markers". We have also rephrased the sentences you mentioned for clarity and coherence.
- b. In the section discussing mtDNA repair, redundancies and awkward phrasings, such as “role of FUS role” and the repetitive definition of SSB, have been revised for precision and conciseness.
- c. Your observation regarding the interpretation of lower DNA repair efficiency is valued, and the text has been modified to reflect a more accurate representation of the relationship between DNA repair and susceptibility to DNA damage.
- d. In Fig. 6, the necessary corrections, such as changing "ntDNA" to "mtDNA" and refining the description of the diagram, have been corrected.

The manuscript has undergone thorough proofreading and editing to enhance its readability and overall presentation. We are grateful for your constructive feedback, which has contributed to improving the quality of our work.

Reviewer #3

Overall Comments:

Kodavati and colleagues aim to elucidate the role of FUS in mediating mitochondrial DNA damage repair and highlight implications in the development of ALS. Using various cell lines, mouse models, and ALS patient samples, the authors describe that endogenous FUS interacts with mitochondrial localized Lig3 and recruits Lig3 to sites of mtDNA damage to effect DNA repair, particularly in settings of oxidative damage. They also describe that ALS-related FUS mutants impair mtLig3 activity, leading to increased mtDNA damage, which can be restored by overexpression of a mitochondria-targeted Lig1. The authors conclude FUS plays a critical role in mtDNA repair and implicate mtDNA damage in the development of FUS-associated neurodegeneration.

The authors build upon their previous work implicating a role for FUS and Lig3 interactions in nuclear DNA repair. Here, they extend their results to show that FUS now binds to the mt-localized isoform of Lig3, this interaction appears to be impacted by ALS-related FUS mutations, and the FUS-mtLig3 complex is distinct from partners found at FUS-Lig3 sites within the nucleus. Further, mtLig3 is less abundant following FUS deletion. Overall, this study provides novel and intriguing insights into the role of FUS in maintaining

mitochondrial genome integrity in neurodegeneration. There are notable areas of improvement for this study detailed below that do not detract from enthusiasm for the work and that if corrected, would strongly improve the rigor and description of the data herein, as well as some minor comments to improve the clarity of the work.

Response: We are grateful for your thoughtful comments and your acknowledgment of the potential impact of our work. After a thorough review, we have addressed all of your concerns and made necessary revisions to the manuscript. Your constructive feedback and suggestions played a crucial role in enhancing the clarity, depth, and overall impact of the content. Thank you for giving us the chance to respond to your concerns.

Major Comments:

Comment-1: The authors describe reduced mtDNA/nuclear DNA ratios for the P525L FUS mutation in Figure S3. Reductions in mtDNA would seem the most important consequence of impaired mtDNA repair that would ultimately explain reduced respiration, yet these data are not more prominently featured. Further, mtDNA/nDNA ratios are not reduced in FUS knockouts. It would be helpful if the authors showed these data in a more useful format (the averaged mean WT and mutants in all conditions side by side on the same graph, including in Figures S3D and S3E).

Response: We appreciate your insightful observation regarding the presentation of our mtDNA/nuclear DNA ratio data, particularly concerning the P525L FUS mutation. In light of your suggestion, we have revised Supplementary Fig. 3 in our manuscript. This revised presentation allows for direct comparison of the differences in mtDNA/nDNA ratios between WT and mutant FUS cells, as well as the effects in FUS knockout conditions. As Ligase is also involved in mitochondrial DNA replication, in addition to repair, mutant FUS-induced Ligase 3 defects could result in reductions in mitochondrial DNA. This possibility has been discussed in the revised manuscript. You need to be careful here- according to the Clayton model of mitochondrial DNA replication (ie heavy and light strand from different origins) you don't need to join Okazaki fragments so there are relatively few if any ligation events needed to replicate the circular mitochondrial genome

Comment-2: As a corollary to the above, mtDNA damage should also lead to reduced mtRNA expression and mitochondrial encoded proteins, beyond reduced mtDNA content. Again, these should be measured and shown to better explain the functional respiratory defects found in FUS mutants.

Response: Thank you for your detailed feedback. We agree that assessing the impact of mtDNA damage on mitochondrial-encoded proteins is crucial for a comprehensive understanding of the potential consequences of respiratory defects in FUS mutants. In response to your suggestion, we have conducted additional analyses to measure the protein levels of three mitochondrially encoded proteins, ND4, and CYTB. These findings have been incorporated into Supplementary Fig. 4d. This additional data enriches our study by providing a clearer link between mtDNA damage and the resultant effects on mitochondrial protein expression, thereby enhancing the overall rigor and depth of our findings. Thank you for guiding us to strengthen this aspect of our research.

Comment-3: There currently is a heavy reliance on 293T mutant FUS cells and iPS-derived fibroblasts for many endpoints in the study which limit the impact of the work, as ALS manifests in neurons. 293T cells and fibroblasts are notorious for being far less dependent on mitochondrial function for cell health compared to neuronal cells. The authors should

perform confirmatory studies of their work (such as functional studies in Figure 2, Lig3 activity assays in Figure 4, and rescue studies with Lig1 in Figure 5) in neuronal cell types.

Response: Thank you for your constructive comment. As we stated in our response to Reviewer-1's comments, we agree that validating key findings in ALS-relevant neuronal cells is crucial for enhancing the clinical relevance and overall quality of the manuscript. Initially, due to the practical challenges associated with scaling up neuronal cultures, we opted to perform experiments necessitating a substantial volume of cells in fibroblasts, reserving procedures such as immunofluorescence or LA-PCR for motor neurons. In response both your and Reviewer-1's suggestion, we have now included additional experiments using ALS-relevant neurons, thus complementing our findings obtained from patient-derived fibroblast cells. These new data include PLA analyses in iPSC-derived motor neurons (Fig. 2e) and immunoblotting conducted from motor neuron extracts (Fig. 2b). In addition, we have conducted experiments using neuronal lineage progenitor stem cells (NPSCs) differentiated from ALS patient iPSC lines in the revised manuscript. These include validation of the functional studies in Fig. 2, Lig3 activity assays in Fig. 4, and rescue studies with Lig1 in Fig. 5. By incorporating these results into our revised manuscript, we aim to provide a more relevant and robust set of data that mirrors the cellular context of ALS more accurately. Your suggestion has been instrumental in elevating the scientific rigor and relevance of our findings, and we are thankful for the opportunity to strengthen our research in this way.

Comment-4: Overall, Figure 3 was a challenge to review. Layouts of all figures were varied throughout the panel and tough to follow (F appears above E, for instance). It would help if this figure was laid out more clearly or split into more figures.

a. The text describing Figure 3E states "The results showed that mtDNA integrity increased significantly at 180 minutes in FUS WT cells, indicating successful DNA damage repair, while repair capacity was reduced in FUS KO cells (Fig. 3e)". This interpretation is misleading. The integrity appears to recover in both groups, while KOs appear worse because of a stronger decline in integrity at Mt LA at 0.5 h. This would suggest the KO is more susceptible to damage rather than unable to repair normally.

b. Further, it would help if the samples in 3E and F were directly compared (Y axes between these studies and graphs are shown in different formats) as it appears that the FUS ALS mutants have worsened recovery than the FUS mutants. Could this suggest a dominant negative effect of these mutants?

c. Figure 3G is a very confusing experiment as it is unclear what is the control group (is "patient tissue the control?") and the layout and Y-axes are very different for each tissue type. There are also no error bars shown. How many times were these experiments performed?

Response: Thank you for your constructive feedback on our manuscript, specifically regarding Fig. 3. We appreciate the opportunity to clarify and improve the presentation of these findings.

a. We acknowledge your concern regarding the interpretation of Fig. 3e and agree that the wording may have unintentionally led to confusion. In our revised manuscript, we have modified the text to more accurately reflect the data: "While mtDNA integrity does recover in both FUS WT and FUS KO cells, loss of FUS showed reduced recovery at the 2 hour time point. One contributing factor for this observed difference may be due to a more pronounced initial decline in integrity in FUS KO cells at 0.5 hour, suggesting an increased susceptibility to damage in these cells, in addition to possible impaired repair." This rephrasing should clarify the observation that the KO cells are more susceptible to initial damage in addition to exhibiting impaired repair.

b. In response to your suggestion for a direct comparison between the samples in Fig. 3e and f, we have revised the figure to standardize the Y-axes, facilitating a more straightforward comparison. This revision should address the concern about differing recovery rates between FUS ALS mutants and FUS mutants. Additionally, we have included a discussion on the potential dominant negative effect of these mutants, providing a deeper insight into the observed variances.

c. We realize that Fig. 3g was not as clear as it should have been. As can be seen from supplementary table 3, the mutation number varied among samples, we used pooled samples for sequencing and sequencing was only performed once.

We believe these revisions will enhance the clarity of our results, and we thank you for your valuable insights that have guided these improvements.

Comment-5: For the FUS proximity ligation assays, several missing controls detract from the results. FUS PLA should be performed on FUS KO cells as a true negative control to ensure non-specific signals are not observed in the absence of one of the interacting proteins. A cytoplasmic or membrane counterstain/marker should also be included to ensure that PLA signals truly are intracellular and do not arise non-specifically elsewhere on the slide. Additionally, the Methods section does not indicate if Z stack images were captured. Imaging Z-stacks would ensure a more accurate representation of PLA signal in 3 dimensions on the imaged cells.

Response: Thank you. We have now included FUS KO cells as a negative control in Supplementary Fig. 1d to address non-specific signals. Furthermore, we have added images co-stained with Mito Tracker, alongside DAPI, to ensure the intracellular localization of the PLA signal. Additionally, we have revised the Methods section to include details about capturing Z-stack images, which will provide a more accurate representation of the PLA signals in three dimensions.

Comment-6: Population sizes used in each group and the specific statistical tests applied should appear in the figure legend for ease of the reader. Additionally, please include the specific data points that comprise the mean values in the bar graphs (as well as both upper and lower error bars). The appearance of the current data is rather opaque as currently shown.

Response: We have revised the figure legends to include comprehensive statistical information and 'n' numbers for each group. Additionally, we have modified the figures to display specific data points comprising the mean values in the bar graphs, along with both upper and lower error bars in relevant figures as per your suggestion.

Comment-7: In Figure 5C, data with use of the mt targeted Lig1 construct is confusing. Following cleavage of the MTS (and the Flag tag), Flag would be expected to diffuse out of the mitochondria. However, Flag is visualized in the mitochondria on the cell fractionation experiment without molecular mass markers (these are missing in all blots in the paper and ideally should be included). Is this full length mt Lig1?

Response: Thank you for your detailed observations regarding the data in Fig. 5C and the use of the mt targeted Lig1 construct. Based on your feedback, we have now incorporated molecular weight markers in all the figures to enhance clarity and accuracy. Regarding the observation of FLAG staining in mitochondria, we have utilized full-length Lig1. We acknowledge our initial oversight regarding the phenomenon of MTS cleavage. After further literature review, we understand that the presence of FLAG in mitochondria could be a transient event, where cleavage occurs during or after import into the mitochondria. This

might explain the FLAG detection in our experiments. It is possible that due to the presence MTS between FLAG and Lig1 sequences, the FLAG-MTS sequence may be hampered. However, if cleavage does occur, then the transient detection of FLAG may not reflect the total Lig1 expression. It is important to note that we have also included total Lig1 levels using Lig1 antibody in addition to FLAG in Fig. 5. However, to avoid confusion and enhance the clarity of our findings, we have removed the FLAG immunoblotting from Fig. 5 and only show total Lig1 levels. Thank you for this critical observation.

Minor Comments:

1. The last part of the title is somewhat misleading. This study does not look at neurodegeneration, but rather the mutations associated with it. Please alter the title to accurately reflect the findings.
2. In most IF images throughout the paper, the scale bar in the bottom right corner appears to be placed on top of another scale bar.
3. Please provide quantitation of Figure 1A showing increase in FUS levels in mitochondria after GO stimulation.
4. The text in the Results describing Figure 1B should mention that there seems to be more P525L FUS in the mitochondria at baseline.
5. Figure 2 is overcrowded and would be easier to read if bar graphs and font were slightly smaller. Group labels are only shown at the bottom of Figure 2E which also was confusing to read.
6. The text states “Given the localization of FUS in mitochondria and the colocalization of mutant FUS in SGs induced by sodium arsenite, we hypothesized that the accumulation of mutant FUS in SGs results in reduced FUS levels in mitochondria, thereby disturbing its normal function”. While the Seahorse did assess function, it did not assess whether FUS levels are reduced in the mitochondria after NaAs treatment.
7. The Figure 2 legend indicates that “Non-mitochondrial oxygen consumption was determined by measuring difference between total oxygen consumption and oligomycin induced reduction in oxygen consumption”. Non-mitochondrial oxygen consumption is generally accepted as the oxygen consumption remaining after (ideally both) Rotenone and Antimycin A are added.
8. The Figure 2 legend indicates that “maximal respiration was assessed following mitochondria uncoupling by FCCP”. Maximal respiratory capacity is generally measured as the difference between oxygen consumption following FCCP and oxygen consumption following Rotenone and Antimycin A. Please clarify.
9. Figure 3 legend states “e shows patient derived fibroblasts whereas f represents HEK 293 cells.” Should this not read that e represents the HEKs and f represents the fibroblasts?
10. In Figures 4A and B, please provide a quantification of Lig3 levels in each of these groups as well as for FUS levels in Figure 5B.
11. In the text “We first tested the overall ligation activity in mitochondria by in vitro ligation activity assay and found that MTS-Lig1 expressing fibroblasts with FUS P525L mutation had a significantly higher ligation activity than the control (Fig. 6d)”. This appears to be mislabeled instead of Figure 5D.

Response: Thank you for pointing out the text-related issues in the manuscript. We have conducted a thorough review to address the typos, grammatical errors, and organizational inconsistencies you highlighted. Specifically:

1. We altered our title, as “**FUS Unveiled in Mitochondrial DNA Repair and Amelioration of Neurodegeneration-associated Mutant FUS-Linked Defects via Targeted Ligase-1 Expression**” to reflect the findings of the publication.
2. We have fixed the representation of scale bars in the images according to your suggestion.
3. The suggested quantification is provided in Fig. 1a.
4. The text has been modified according to the suggestion for Fig. 1b.
5. Fig. 2 has been modified according to the suggestion.
6. We have modified Supplementary Fig. 2d to show FUS level decrease in mitochondria post NaAs treatment.
- 7 and 8. The corrections to the Fig. 2 legends have been made according to the suggestions and have been clarified in the text.
9. The Fig. 3 legend has been revised per the suggestion.
10. Quantifications are provided for Lig3 and FUS levels in Fig. 4 and 5.
11. The mislabeling of Fig. 5d has been corrected.

Reviewer #1 (Remarks to the Author):

None

Reviewer #2 (Remarks to the Author):

The authors have addressed my prior concerns and expanded their characterization of the mtDNA defects in FUS ALS samples, enhancing the significance of their work. I have only a one minor point to address:

1. LA-PCR does not measure strand breaks. It assesses the presence of DNA polymerase amplification blocking lesions. Certainly, these can be strand breaks, but could represent many other DNA lesion types as well. That should be corrected.

Reviewer #3 (Remarks to the Author):

I have no remaining concerns, the manuscript has been nicely revised and has addressed my questions.